# Metallic micronutrients are associated with the structure and function of the soil microbiome

Zhongmin Dai[1,2,3], Xu Guo[1,2], Jiahui Lin[1,2], Xiu Wang[1,2], Dan He[1,2], Rujiong Zeng[1,2], Jun Meng[4], Jipeng Luo[5], Manuel Delgado-Baquerizo [6], Eduardo Moreno-Jiménez [7,8], Philip C. Brookes[1,2] & Jianming Xu [1,2,3] ✉

The relationship between metallic micronutrients and soil microorganisms, and thereby soil functioning, has been little explored. Here, we investigate the relationship between metallic micronutrients (Fe, Mn, Cu, Zn, Mo and Ni) and the abundance, diversity and function of soil microbiomes. In a survey across 180 sites in China, covering a wide range of soil conditions the structure and function of the soil microbiome are highly correlated with metallic micronutrients, especially Fe, followed by Mn, Cu and Zn. These results are robust to controlling for soil pH, which is often reported as the most important predictor of the soil microbiome. An incubation experiment with Fe and Zn additions for five different soil types also shows that increased micronutrient concentration affects microbial community composition and functional genes. In addition, structural equation models indicate that micronutrients positively contribute to the ecosystem productivity, both directly (micronutrient availability to plants) and, to a lesser extent, indirectly (via affecting the microbiome). Our findings highlight the importance of micronutrients in explaining soil microbiome structure and ecosystem functioning.

Micronutrients such as iron (Fe), manganese (Mn), copper (Cu), zinc (Zn), molybdenum (Mo), and nickel (Ni) are critical regulators of microbial-driven processes such as photosynthesis, respiration, biomolecule synthesis, redox homeostasis, and cell growth and immune system functioning[1–3]. Although they are required by organisms in a small amount, the deficiency of micronutrients significantly limits organism growth and biological processes. Soil pH and macronutrients are regarded as the major predictors of the structure and function of

soil microbiome[4,5]. Strikingly, very little is known on how metallic micronutrients correlate with the soil microbiome across environmental gradients.

Micronutrients may help explain soil microbiomes for three main reasons. First, most cells and enzymes associated with microbial reproduction and regulated biological processes require soil micronutrients, e.g. Fe and Mn for microbial respiration, Cu and Zn for immunocompetence, and Fe, Mo and Ni for N fixation, etc. Second,

[1]Institute of Soil and Water Resources and Environmental Science, College of Environmental and Resource Sciences, Zhejiang University, 866 Yuhangtang Road, Hangzhou 310058, China. [2]Zhejiang Provincial Key Laboratory of Agricultural Resources and Environment, Zhejiang University, 866 Yuhangtang Road, Hangzhou 310058, China. [3]The Rural Development Academy at Zhejiang University, Zhejiang University, Hangzhou 310058, China. [4]Zhejiang Province Key Laboratory of Recycling and Ecological Treatment of Waste Biomass, School of Environmental and Natural Resources, Zhejiang University of Science and Technology, Hangzhou 310023, China. [5]Ministry of Education Key Laboratory of Environmental Remediation and Ecological Health, College of Environmental and Resource Sciences, Zhejiang University, Hangzhou 310058, China. [6]Laboratorio de Biodiversidad y Funcionamiento Ecosistémico. Instituto de Recursos Naturales y Agrobiología de Sevilla (IRNAS), CSIC, Av. Reina Mercedes 10, E-41012 Sevilla, Spain. [7]Department of Agricultural and Food Chemistry, Faculty of Sciences, Universidad Autónoma de Madrid, 28049 Madrid, Spain. [8]Institute for Advanced Research in Chemical Sciences, Faculty of Sciences, Universidad Autónoma de Madrid, 28049 Madrid, Spain. ✉e-mail: jmxu@zju.edu.cn

many soil C, N, and S redox reactions are coupled with micronutrient elements that are able to provide and accept electrons (e.g. Fe and Mn)[6–8]. Third, soils evolved from parent materials or secondary minerals that contains different micronutrients produce contrasting habitats, which would favor some microorganisms but not others[9]. Therefore, the micronutrients probably influence all aspects of microbiome such as microbial abundance, diversity, network connectivity (i.e. potential interaction between microorganisms) and associated functions involved in nutrient cycling. Finally, soil pH is regarded as the main driver of soil microbial communities, but it is also a critical influencer of micronutrient availability. In this respect, micronutrients may help to explain the still poorly understood mechanisms behind the well-described relationship between pH and the structure and function of microbiomes.

Microorganisms also form stable mutualistic relationships with plant roots, contributing to plant immunity as well as nutrient uptake[10]. For example, microbial abundance, diversity, and network complexity have been reported to be positively correlated with ecosystem productivity[11–13]. Moreover, microorganisms participate in element cycling such as organic C decomposition, N fixation, nitrification and denitrification, P solubilization, and S sulphidisation[14–16]. These functions influence the carbon sequestration and nutrient availability to plants, also benefiting ecosystem production. Recent studies report that soil micronutrient availability together with soil other properties (e.g. pH, macronutrient, etc) affect ecosystem productivity[17], while whether micronutrients contribute to ecosystem production also by affecting the soil microbiome remains to be tested. Because of this, identifying the role of micronutrients in explaining soil microbiomes is critical to increase our capacity to predict ecosystem responses to environmental changes.

To better understand the relationship between micronutrients and the structure and function of soil microbiome, and how these changes affect the ecosystem production, we conducted a large spatial collection of soil samples from 180 sites in China, with diverse ecosystems including contrasting soil properties, climatic conditions and vegetation types from tropical to cold regions (Fig. S1–3). The mean annual precipitation and temperature range from 395 to 2486 mm and −2.7 to 27.9 °C, and these sites are representative of other parts of the globe. The availability and total amount of Fe, Mn, Cu, Zn, Mo, and Ni, bacterial and fungal abundance, diversity, network connectivity and associated C, N, P and S gene abundances were evaluated. We aimed to examine the relationship between micronutrients and microbial community composition and functioning, further the ecosystem production, and compare the effects with the well-known predictors of soil pH and macronutrients. We hypothesized that micronutrients can explain a unique portion of the variation in soil microbiome structure and function in addition to pH and macronutrients. Our findings highlight that micronutrients are associated with the structure and function of soil microbiomes, and also with the ecosystem production.

## Results

### Microbial total abundance

Micronutrient content was significantly correlated with bacterial abundance, but was not associated with fungal abundance. The available PC1 (with correlation coefficient of 0.27) and total PC2 (0.23) had positive relationships with bacterial total abundance (Fig. S4). Bacterial total abundance was also positively correlated by the available Fe (0.27), Zn (0.28), Cu (0.15) and Ni (0.15), and total Mn (0.16) (Fig. S4). However, the fungal total abundance was not influenced by either available or total principal components (Fig. S4) and was only positively correlated with available Zn (0.18) and Mo (0.17), and total Zn (0.17) (Fig. S4). Soil macronutrients and pH were positively correlated with bacterial total abundance (except soil C:N), while fungal total abundance was only influenced by soil pH and total P (Fig. S5).

### Abundance of specific genera

Micronutrients had both negative and positive correlated taxa at the genus level (Figs. S6 and S7). Available Fe had the largest number of correlated bacterial (i.e. 129) and fungal (i.e. 43) genera compared to the other micronutrients (Fig. S6). For example, the lower abundances of taxa such as *Pir4_lineage* and *Gibberella* and the higher abundances of taxa such as *SHA-26* and *Scolecobasidium* were associated with higher available Fe (Fig. 1 and Table S1). Total Mn had the largest number of the sum of positive and negative correlated bacterial (i.e. 114) and fungal (i.e. 55) genera compared to the other micronutrients (Fig. S7). For example, the lower abundances of taxa such as *FCPS473* and *Saitozyma* and the higher abundances of taxa such as *Promicromonospora* and *Gibberella* were associated with higher total Mn (Fig. 1 and Table S2). Further, the higher abundance of *Thrichoderma* and *Rubrobacter* were associated with lower total Cu and available Zn (Fig. 1, Tables S1 and S2). The higher abundance of *Fusarium* was associated with lower total Mo and higher available Ni (Fig. 1, Tables S1 and S2).

### Microbial diversity and co-occurrence networks

The land use and its interaction with climate affected the microbial community (Fig. S8). Except for these factors, the micronutrient content was significantly correlated with bacterial and fungal communities, and the correlations with bacteria were stronger than those with fungi (Fig. 2). The available PC1 had negative correlations with bacterial alpha diversity, beta diversity, and the abundance of dominant phyla such as *Acidobacteria*, *Actinobacteria* and positive correlations with the abundance of *Proteobacteria* (Fig. 2a). Especially, the available Fe correlated most strongly with most of the above parameters and had a significant relationship with bacterial network connectivity (bacteria have more relationships among themselves when connections increase) shown by the total degree, average degree and average path length. The available Cu, Zn, and Mo negatively correlated with bacterial alpha diversity, beta diversity, the abundance of *Acidobacteria* and positively with the abundance of *Proteobacteria*. The number of correlated parameters with total micronutrient contents was fewer than with their available forms. Soil pH, C:P and N:P ratios significantly affected the bacterial diversity, dominant phyla abundance, and bacterial network connectivity (Fig. S9). In contrast, fungal parameters were barely correlated with soil micronutrients, regardless of the soil total or available pools, except for fungal network connectivity and beta diversity (Fig. 2b). Soil pH, C:P and N:P ratios affected fungal beta diversity and network connectivity (Fig. S9).

### Microbial functional genes involved in C, N, P, and S cycling

Micronutrients, together with soil pH and macronutrients, greatly affected the microbial gene abundance in C, N, P, and S cycling (Fig. 3 and Table S3). The available PC1 (with the number of 48 correlated genes), available Fe (42) and available Zn (60) had the largest number of correlated genes, followed by total Mn (24), available Cu (17), total PC2 (11), available Mo (9), total Zn (5) and available Ni (4) (Fig. 3a). The number of genes which were correlated with soil macronutrients and pH was slightly larger than with micronutrients (Fig. 3b).

Specific micronutrients also positively affected the microbial-regulated nutrient pathways. Available Fe was associated with both starch and cellulose degradation by positively correlating with the *amyX* gene and *cdh* gene. Total Mn was significantly positively correlated with the *manB*, *mpn* and *pox* genes that were responsible for the degradation of hemicellulose and lignin, respectively (Fig. 3c). Available Fe, Cu, and Mo affected the microbial denitrification process by correlating with the *narG* and *nirS/K* genes. The Zn significantly correlated with the *napA* and *nirK* genes responsible for nitrogen reduction, *amoA* gene for nitrification and the *bpp* gene responsible for organic P mineralization (Fig. 3d, e). The N$_2$ fixation was influenced by the available Fe (Fig. 3d), while available Fe had no correlations on

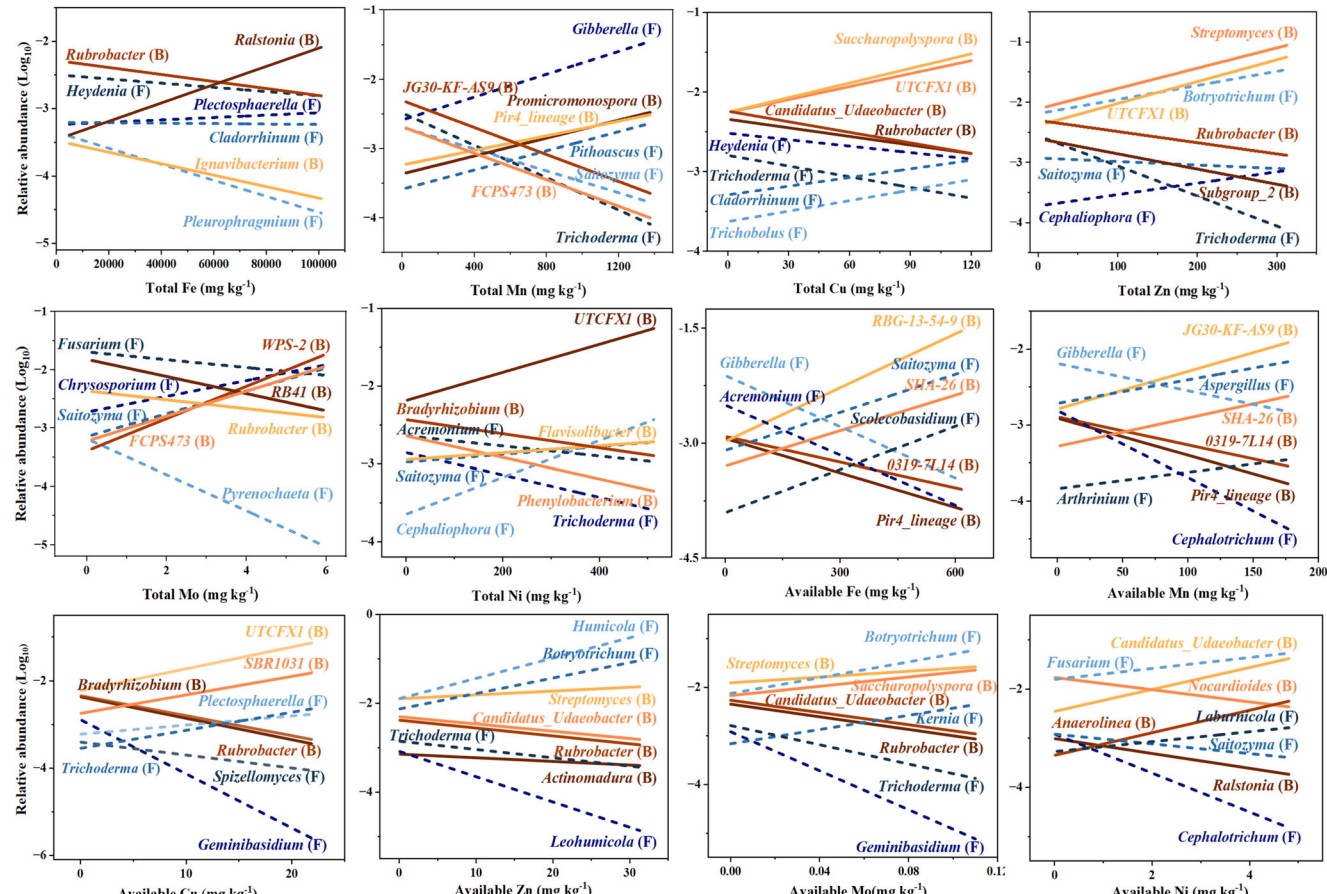

**Fig. 1 | Effects of total and available micronutrients (Fe, Mn, Cu, Zn, Mo, and Ni) on the abundance of soil microbial taxa.** Microorganisms at the genus level were identified by Deseq2, having relative abundances that were significantly higher or lower in relatively low micronutrients as compared with the high micronutrient group (Figs. S6 and S7). Two negatively and positively identified genera with the most significance (i.e. lowest adjusted p-value) are presented in the Figure. The solid and dashed lines marked with "B" and "F" present the bacterial and fungal genera respectively. The correlation coefficients with p-value adjustment are presented in Tables S1 and S2.

P transformation processes (Fig. 3e). Available Fe and Cu also affected the microbial S reduction process by correlated with the *dsrA* and *dsrB* genes (Fig. 3f). Notably, the *Ralstonia* (with the average relative abundance of 0.4% in a whole community) that correlated with Fe also contained the Fe-responsive genes such as *amyX* and *narG* (Table S4). The *streptomyces* that correlated with Zn (2.3%) contained the Zn-responsive genes such as *napA*, *nirK* and *bbp* (Table S4).

## Effects of Fe and Zn addition on soil microbiome

The Fe and Zn additions greatly affected the structure and function of the soil microbiome in the incubation experiments. The Fe addition significantly decreased the Shannon index and changed bacterial community composition in most soils, except for the soil from GD site (Fig. 4a), which was consistent with negative correlation between available Fe and Shannon index/PCoA1 from the observational study (Fig. 2a). Similar finding of alpha diversity and community composition was observed in Zn addition treatments (Fig. 4b), in agreement with the negative correlation between Shannon index/PCoA1/PCoA2 (Fig. 2a). Similar to the positive correlation with the relative abundance of *Proteobacteria* (Fig. 2a), the Fe and Zn additions increased the relative abundance of *Proteobacteria* in most soils (Fig. 4). With microbial function, the Fe addition increased the relative abundances of *cdh*, *nifH*, *dsrB* and *nirS* genes in the soils from NM and YN, while had no increasing effects on the gene abundances in the soils from GD and JS, both soils having higher initial Fe concentration than NM and YN

(Fig. 4a). The Zn addition increased the relative abundances of *bbp* gene in the soils from GD, SD and YN sites, and also increased the gene abundance for *amoA gene* in the soils from GD, NM and YN sites (Fig. 4b). The increases in these gene abundances were consistent with their positive correlations with available Fe/Zn concentrations from the observational study (Fig. 3).

## Contribution of micronutrients to ecosystem production

Micronutrients contributed to ecosystem production by both direct and indirect effects (Fig. 5). With the bacterial model, ecosystem production was co-correlated by soil pH, bacterial abundance, and micronutrients. Soil micronutrients affected the ecosystem production mainly by direct effects and also by increasing bacterial abundance (Fig. 5a). The micronutrient-affected bacterial diversity and network connectivity were not correlated with the ecosystem production (Fig. 5a). With the fungal model, ecosystem production was co-correlated by soil pH, total C, fungal abundance and micronutrients. Soil micronutrients directly and positively correlated with the ecosystem production, but not by affecting fungal abundance, diversity, and network connectivity (Fig. 5b). With the function model, the ecosystem production was co-correlated by soil pH, micronutrients and microbial functions involved in C and N cycling. However, micronutrients did not affect ecosystem production by altering microbial function. Instead, soil pH and total C that positively correlated with the microbial functions contributed to the ecosystem production (Fig. 5c).

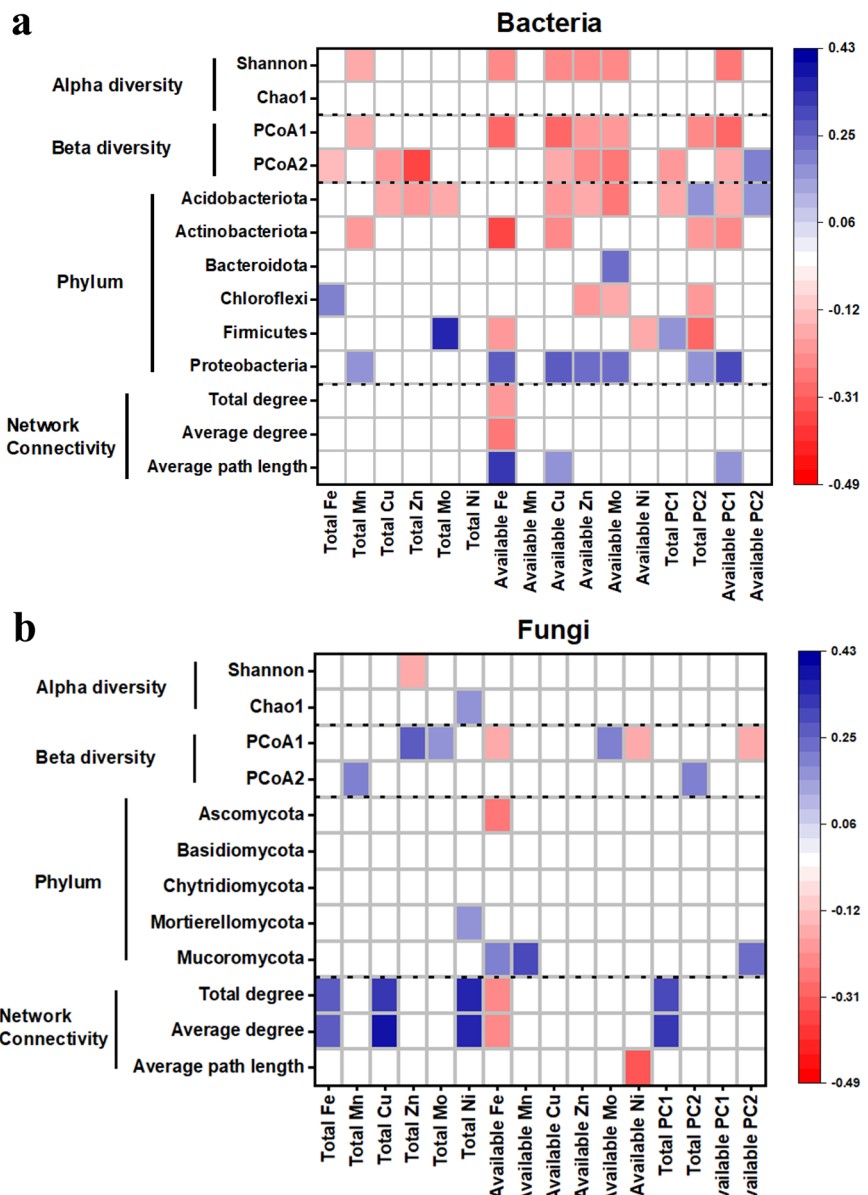

**Fig. 2 | Effects of total and available micronutrients (Fe, Mn, Cu, Zn, Mo, Ni) on soil microbial communities. a** Heatmap shows the relationship between alpha, beta diversity, phylum abundance, and network connectivity for bacteria and micronutrients, conducted by partial correlation. **b** Heatmap shows the relationship between alpha, beta diversity, phylum abundance, and network connectivity for fungi and micronutrients, conducted by partial correlation.

## Discussion

The influence of micronutrients as main regulators of microbial communities is being increasingly acknowledged in marine ecosystems[18] and in the human gut microbiome[19]. Yet, the contribution of micronutrients to explain the structure and function of soil microbiomes in terrestrial ecosystems remains largely understudied.

Our findings, based on an observational study across China and a laboratory validation, highlight the importance of soil micronutrients in explaining the diversity and community composition of soil microbes. Theoretically, micronutrients participate in the auxiliary metabolism and soil redox reaction, and the minerals containing micronutrients create different habitats. These mechanisms would create a more specialized microbial community, causing overall diversity and community composition changes. Previous studies have reported that micronutrients explained a large variation in the structure of microbial communities in agricultural soils receiving specific micronutrients[20], while we further confirm the relationship between micronutrients and microbial structure in a wider range of soils with more contrasting properties and under different land uses. Our finding is also consistent with some experiments showing that iron minerals shaped soil microbial community[21] and Zn fertilization decreased soil microbial diversity and altered community composition in paddy soil[22]. In general, soil pH is previously reported to be positively linked to soil bacterial diversity[4,23] and also negatively linked to micronutrient availability[24]. The relationship between micronutrients and microbial diversity can be masked by the pH effects. Thus, our analyses controlled the pH effects on the soil microbiome statistically and provided solid evidence that the micronutrients, especially their availability, can help explain microbial diversity (Fig. 2). The direct strong relationship between micronutrients and the structure of the soil microbiome can partly explain previous evidence of enrichment of specific microorganisms in Fe-Mn nodules[25] and the high Fe associated-microbial occurrence[26]. It may also assist in helping to explain the unknown driven factors in bacterial communities where the soil pH and

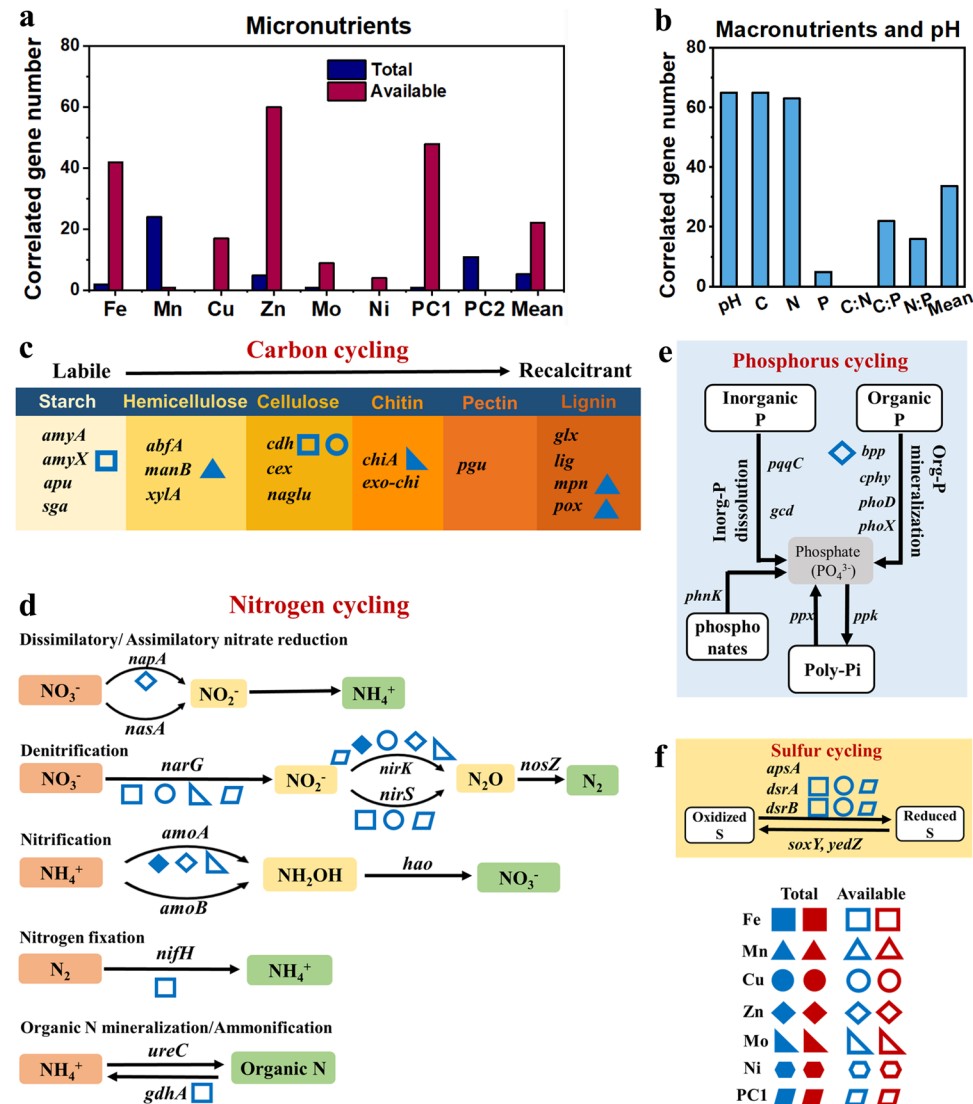

**Fig. 3 | Effects of total and available micronutrients (Fe, Mn, Cu, Zn, Mo, Ni) on soil microbial regulated functions involved in C, N, P, and S cycles. a** Number of genes involved in microbial C, N, P, and S cycles whose absolute abundances significantly correlated with soil micronutrients by partial correlation. **b** Number of genes whose absolute abundances significantly correlated with soil chemical properties by partial correlation. The "mean" represented the average number of genes correlated with all micronutrients or soil macronutrient+pH.
**c** Micronutrient-participated microbial pathways for the C cycling. **d** Micronutrient-

participated microbial pathways for the N cycling. **e** Micronutrient-participated microbial pathways for the P cycling. **f** Micronutrient-participated microbial pathways for the S cycling. The labels of micronutrients presented in the pathways were strictly filtered if their correlations with the absolute abundance and relative abundance of corresponding genes were both significant ($p < 0.05$). Labels with red and blue colors presented that negative and positive correlations between micronutrients and gene abundance. The microorganisms containing the genes in the pathways at the phylum and genus level are presented in Table S4.

macronutrients are considered[27]. Outstandingly, the fungal community (e.g. abundance and diversity) was less sensitive to soil micronutrients compared to bacteria. Being both heterotrophic and eukaryotic organisms, they are probably more dependent on energy resources (C and N) rather than micronutrients and more resistant to environmental changes caused by micronutrient addition than bacteria[5].

Our study also extended the field study to gene abundance and expected soil functions, on the basis of microbiome structure[20]. We revealed that micronutrients were correlated with the abundance of microbial functional genes associated with the C, N, P, and S cycle. Coupled reactions of macro and micronutrients are expected to be essential to support soil macronutrient turnover, flux and pool sizes, and nutrient availability to plants (e.g. labile C, $NO_3^-$, etc.). Specifically, one major finding shows micronutrients, especially Fe and Mn, participated in microbial-regulated soil nutrient cycling, such as organic C

decomposition, C fixation, denitrification, S reduction, and potentially influencing ecosystem functioning (Fig. 3). Micronutrients such as Fe and Mn usually participate in microbially-regulated redox reactions of C, N and S in soils[8], because micronutrients are involved in electron availability and transfer in biochemical reactions[6,28]. For instance, soil Fe and Mn speciation change from reduced forms under anaerobic conditions to oxidized forms under aerobic conditions, and this couples with C and N transformations[29], while sulfate-reducing bacteria are involved in Fe and Mn oxidation[30]. In addition, micronutrients such as Cu and Zn positively explained some microbial pathways of soil organic C decomposition, nitrification, and denitrification that are related to soil respiration and greenhouse gas emissions (e.g. $CO_2$ and $N_2O$). This indicates the potential influence of these micronutrients on soil-climate change feedback that should be further evaluated[2,23,31]. Another important finding is that the micronutrients always positively correlated with gene abundances (Fig. 3). Over the moderate range of

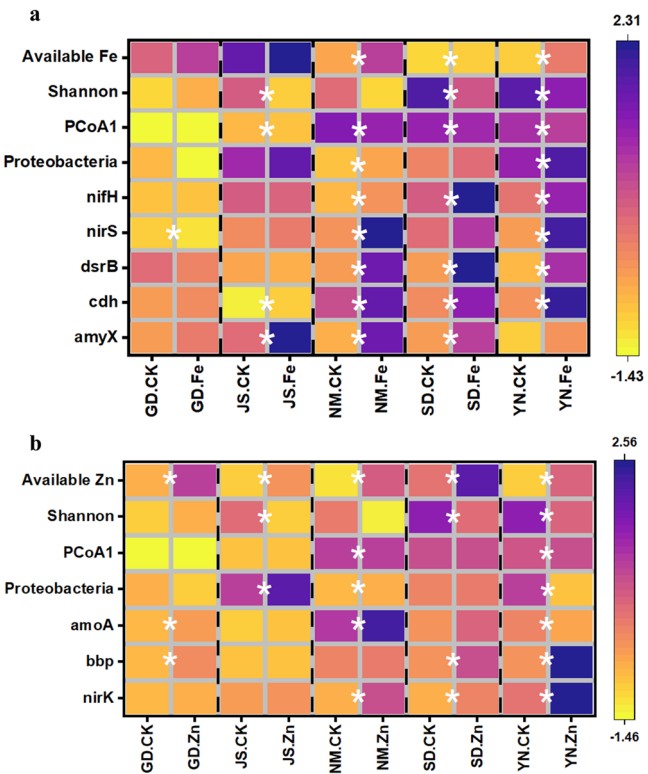

**Fig. 4 | Effects of Fe and Zn additions on soil micronutrient availability and microbial parameters in soil incubation experiments. a** Changes in soil available Fe concentration, microbial diversity, *Proteobacteria,* and relative abundances of *nifH*, *nirS*, *dsrB*, *cdh*, and *amyX* after Fe addition. **b** Changes in the soil available Zn concentration, microbial diversity, *Proteobacteria* and relative abundances of *amoA*, *bbp* and *nirK* after Zn addition. The differences in microbial parameters normalized by Z-score between control and the Fe/Zn treatments were conducted by independent t-test. The "*" represents the differences were significant at p < 0.05.

metals in our soils (i.e. non-contaminated soils), the supply of micronutrients may not cause toxicity or inhibition of microbial growth in any case. Instead, they will enhance microbial growth and enzymatic activities, indicating the moderate increase of micronutrients can accelerate the macronutrient cycling in terrestrial ecosystems. Our incubation experiments also revealed the beneficial impacts of micronutrients Fe and Zn on microbial-specific functions. The added Fe and Zn demonstrated the potential to serve as coenzymes or induce alterations in soil conditions[6,32]. However, the Fe and Zn effects depended on soil types. For example, the Fe addition in the soil from GD did not change microbial functions (Fig. 4), probably attributed to the high initial Fe content in red soil. On the other hand, certain microbial functions might exclusively exhibit changes within specific incubation conditions, such as under highly anaerobic conditions or in the presence of other coexisting elements.

Further analyzes identified both bacterial and fungal genera that were highly correlated to different micronutrients, e.g. *O319-7L14* to Fe, *Aspergillus* to Mn, and *Fusarium* to Mo and Ni that have much lower concentrations in soils (Fig. 1). As the relationship for most identified genera have not been previously reported, these genera can be used as biomarkers predicting the specific habitats having one extremely high or low micronutrient. In addition, some Fe positively-correlated genera would be potential magnetotactic microorganisms that predict the high content of magnetite, goethite, or ferrihydrate in soil[33]. Some genera participated in important soil processes such as Fe-correlated starch decomposition (regulated by *Microlunatus*), Zn-correlated nitrification (*Candidatus Nitrosocosmicus*), and denitrification (*Solirubrobacter*),

Zn-correlated organic P mineralization (*Solirubrobacter*), and Cu-correlated S reduction (*Piscinibacter*) (Table S4). The identified microorganisms will be important microbial resources for regulating soil nutrient cycling and maintaining ecological functions, especially in micronutrient-abundant habitats.

We further demonstrated that the contribution of micronutrients to microbial communities depends on the availability of micronutrients. Regarding micronutrient pools of biological significance, total stocks are more stable over time, while the available pools are more mobile and soluble in soils and expected to be more accessible to microorganisms[9]. Thus, in our study, microbial diversity and the abundance of specific phyla were more correlated with micronutrient availability than its total stocks (except for Mn and Mo). Also, some micronutrients are present in soils in biologically inert solid phases. For instance, soil total Fe accounts for more than 3 % of soil mass but most of it is fixed as iron oxides or other minerals[34], that are included in the total fraction. Only a small part of it is readily bio-available. In contrast, total Mn and Mo had relationships with more microbial parameters relative to their available form and we attributed it to the more comparable order of magnitudes in total concentration and the respective available concentration for both elements. Moreover, we found that Fe had the largest number of correlations with microbial parameters, followed by Mn, Zn, and Cu, and Mo and Ni had the lowest number. This was partly expected because previous work suggests the role of Fe, Mn, Cu, and Zn in many important enzymatic processes in soils and act as reactants for C, N, and S coupled reactions, while Mo and Ni are only specialized to less processes[6,29,31,35]. Although each single micronutrient plays specific biological roles, the integrated effects of micronutrients (i.e. revealed by PCA) explained more variances for bacterial abundance, diversity and functional genes compared to the specific effect of each micronutrient (Fig. 2a and Fig. 3a).

Recent studies showed that plant productivity is not only influenced by climate and edaphic factors, but also is highly associated with belowground microbial communities (for example, globally, 64% of plant biomass is promoted by soil microbiome restoration)[13]. Given that micronutrients are correlated with microbial community and function, we speculate that micronutrient-driven microbiomes contribute to ecosystem production. Our SEM model revealed that soil micronutrients explain ecosystem production mainly by a direct effect, probably supplementing micronutrients for plant growth[17,36]. However, the SEM model also suggests that higher micronutrient concentrations benefit plant productivity by increasing soil bacterial biomass, shown by the high positive relationship between bacterial biomass and plant productivity (Fig. 5). This is probably associated with the evidence that microorganisms can increase plant nutrient acquisition and resistance to stresses[10]. However, the micronutrient-driven microbial diversity and network connectivity (i.e. indirect effect) did not contribute to ecosystem production, suggesting that the effects of micronutrient-driven microbiome structure on ecosystem production should not be overstated. Micronutrients-drive function also did not contribute to the ecosystem. This is probably attributed to the much lower concentrations of micronutrients compared to macronutrients in soil. For example, the contribution of microbiome function (e.g. C and N cycles) driven by total C to ecosystem production was more dominant (Fig. 5). The increased C cycling may increase the soil C fluxes and organic C loss, therefore negatively affecting ecosystem production. While the increased N cycle may lead to more inorganic N released to the environment, therefore increasing plant growth. Overall, it is important to consider all these factors together to explain ecosystem production including macronutrients, soil pH, and spatial and climatic factors[17,37].

In conclusion, micronutrients are associated with the structure and function of soil microbiomes, highlighting the importance of micronutrients on ecosystem functioning (Fig. 6). The effects of

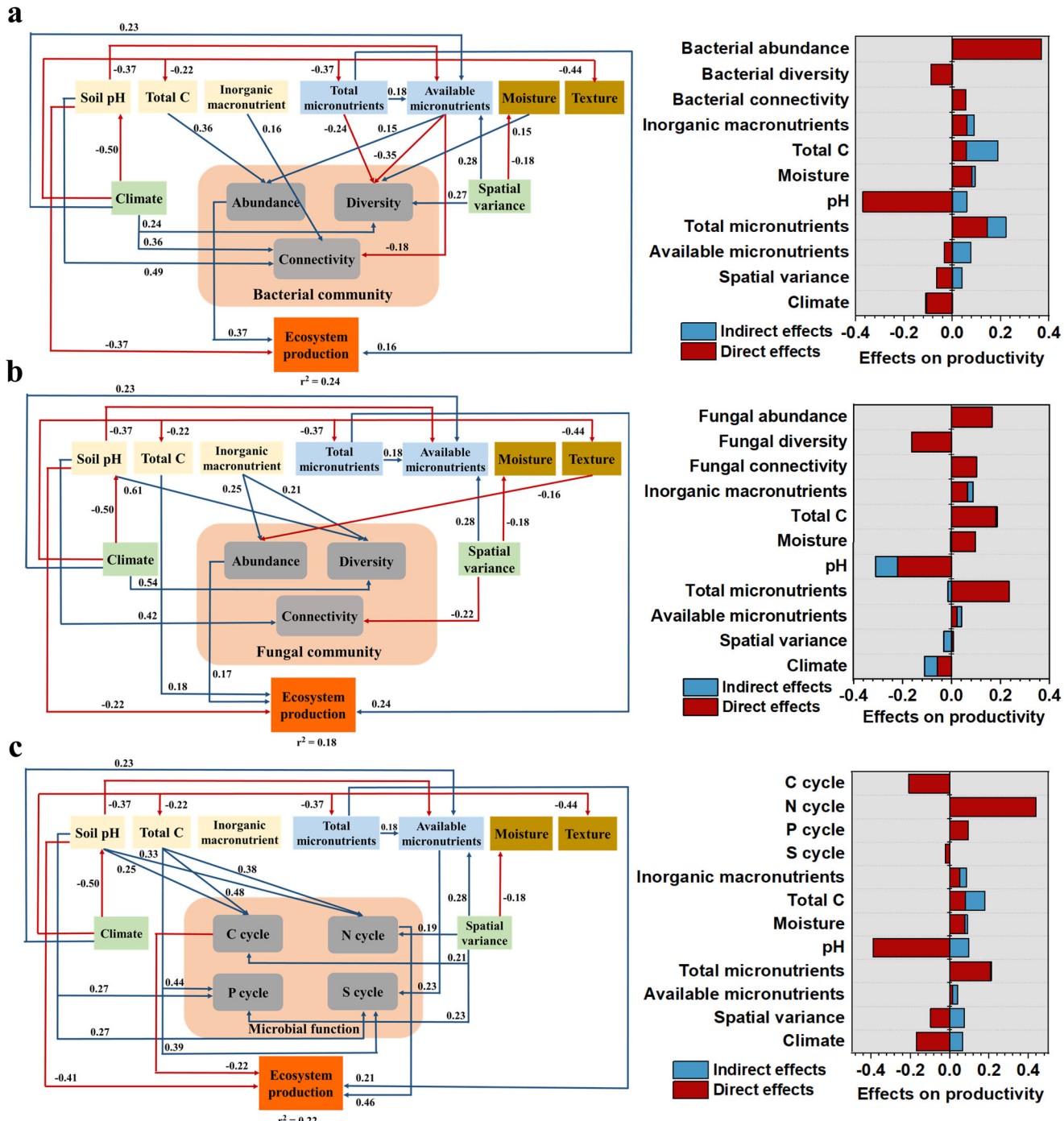

**Fig. 5 | Contributions of environmental variables to ecosystem productivity.**
**a** Effects of environmental variables (i.e. climate, geo-spatial variance, soil basic properties, soil macronutrients, and soil micronutrients) on ecosystem production via altering bacterial community. **b** Effects of environmental variables on ecosystem production via altering fungal community. **c** Effects of environmental variables on ecosystem production via microbial functions. Bacterial community and fungal community include the abundance, diversity, and network connectivity parameters. Microbial function includes the abundance of microbial genes involved in C, N, P, and S cycling. The red and blue arrows represented the significant ($p < 0.05$) negative and positive relationships between variables respectively, while the arrows with non-significant relationships were not shown. Adjacent values near the arrows indicate the path coefficients. $r^2$ value indicates the proportion of ecosystem production explained by each variable.

micronutrients on microbial communities were larger for bacteria than for fungi, with the soil-available pool of micronutrients more strongly associated than the total pool. Notably, the Fe correlated most with the microbial community compared to Mn, Cu, Zn, Mo, and Ni. While the role of micronutrients should not be overestimated, our findings suggest that it should be taken into account in future research and global sustainable development goals.

## Methods

### Site description and sample collection

Surface soils were collected from 180 sites across China, with a span of 32.5° longitude and 33.4° latitude. These locations vary with different climatic conditions (from cold to tropical zones), soil conditions, and vegetation cover (crops, grass, and forests). The mean annual precipitation and temperature ranges from 395 to 2486 mm and −2.7 to

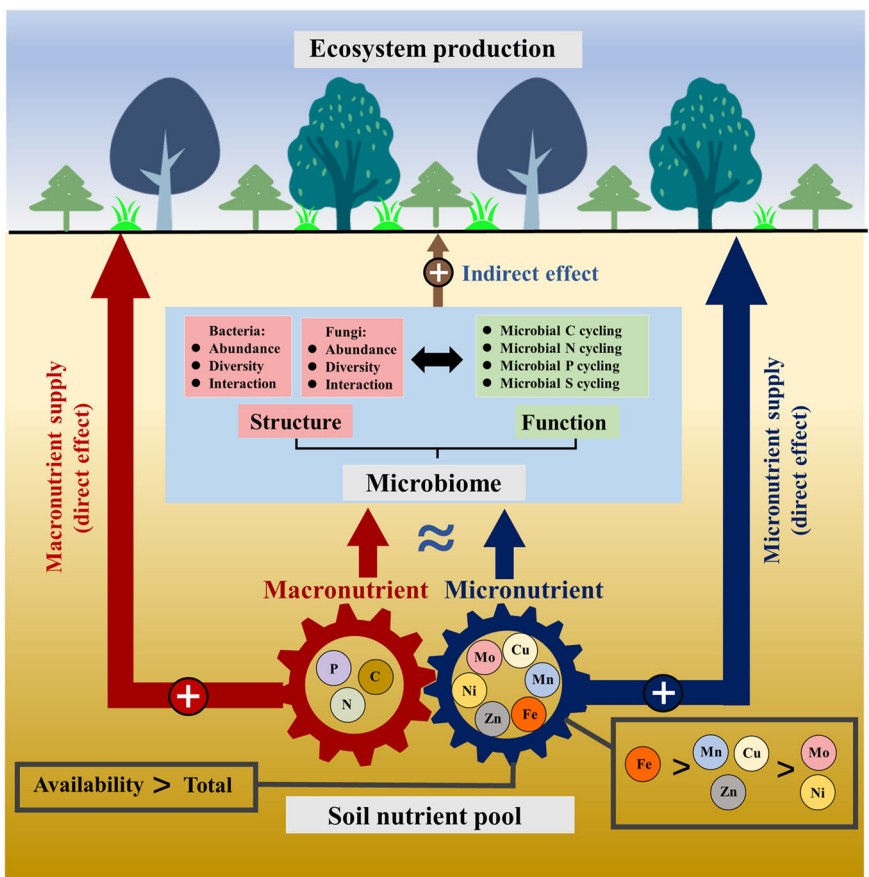

**Fig. 6 | Conceptual diagram illustrating the impacts of soil micronutrients and soil macronutrient on the structure and function of soil microbiome.** Micronutrients are highly correlated to the structure and function of soil microbiomes, and are comparable to the effects of soil macronutrients and pH. Micronutrients positively contributed to the ecosystem productivity by direct effect (nutrient supply for plants) rather than indirect effect (associated microbiome). The icons referring to vegetation used in the figure were applied without endorsement from the website of https://www.iconfont.cn/.

27.9 °C, respectively. The NDVI value ranged 120 from 0.22 to 0.90 across sites. The sample distribution and associated information such as land use are presented in Fig. S1.

Surface soils (0–20 cm), i.e. bulk soils, from each site were sampled. Sample sites were chosen in a central area with the same land use to avoid the disturbance of other land uses and certain features (e.g. roads, buildings). In each site, five soil cores were collected and then pooled to form a composite sample within a size of 15 m x 15 m. The soil samples were immediately transported to the laboratory on ice. The soils were then sieved through 2 mm after the removal of plant residues and stones. Each soil sample was divided into two parts; one was air-dried, ball-milled, sieved to <0.15 mm, and stored at 4 °C before chemical analyzes, and the other was stored at −80 °C prior to DNA extraction. All soil samples were synchronously analyzed for chemical properties and DNA extraction.

### Soil chemical property characterization

Soil pH was measured with a pH meter in a soil-to-water suspension of 1:2.5. Soil total carbon (C) and nitrogen (N) were determined by an elemental analyzer (Vario EL II, Germany). The molybdate blue colorimetric method was used to measure soil total P after the digestion of the soil samples with $H_2SO_4$–$HClO_4$. Soil pH and total C content range from 4.4 to 9.9 and 0.3% to 7.4%, respectively. The details of the range of soil pH and macronutrient contents across the 180 soil samples are presented in Fig. S2.

Soil total concentration of metallic micronutrients including Fe, Mn, Cu, Zn, Mo, and Ni were determined after digestion with

$HF$-$HNO_3$·$H_2O_2$ (1:2:1, v/v/v) at 210 °C for 60 min using a microwave[38]. All the digestion solutions were filtered through filter papers and then adjusted to a volume of 50 mL with distilled water. Soil available micronutrients were extracted by an extractant containing 0.005 mol/L DTPA, 0.01 mol/L $CaCl_2$, and 0.1 mol/L triethanolamine at a pH of 7.30 and the extraction solutions were filtered through a grade 42 Whatman filter paper[39]. This extraction has been commonly used for different soils during the last decades. The concentrations of total and available micronutrients were measured by inductively coupled plasma optical emission spectrometry (ICP-OES) (iCCP 6300, Thermo Scientific, USA). The range of total and available micronutrient concentrations across 180 soil samples are presented in Fig. S3.

### Microbial DNA extraction and sequencing

Soil microbial DNA was extracted by the FastDNA SPIN kit (MP Biomedicals, Solon, OH, USA) following the manufacturer's protocol. Soil DNA samples were then sent to for library preparation and amplicon sequencing after quality checked by agarose gel electrophoresis. The primers of 515F-907R (5′-GTGCCAGCMGCCGCGGTAA-3′, 5′-CCGTC AATTCCTTTGAGTTT-3′) were targeted for the 16 S V4 version of soil bacteria[40], and the primers of ITS3F-ITS4R (5′-GCATCGATGAAGAA CGCAGC-3′, 5′-TCCTCCGCTTATTGATATGC-3′) were targeted for the ITS2 version of soil fungi[41]. PCR reactions were prepared as follows: a total volume of 50 μl reaction mixture containing 25 μl of 2x Premix Taq (obtained from Takara Biotechnology, Dalian Co. Ltd., China), 1 ul of each primer (10 μmol/L), and 3 μl of DNA. The thermal cycling program consisted of: 1) an initial denaturation at 94 °C for 5 min, 2) 30

cycles of denaturation at 94 °C for 30 s, annealing at 52 °C for 30 s, and extension at 72 °C for 30 s and 3) final elongation at 72 °C for 10 min. The PCR amplification was carried out using the BioRad S1000 PCR instrument (Bio-Rad Laboratory, CA, USA). Amplified PCR products were sequenced on the Illumina paired-end platform with the instrument of NovaSeq 6000 (Illumina, San Diego, CA, USA). The quality of raw data was controlled by the Fastp (https://github.com/OpenGene/fastp, version 0.20.0) with sliding window (-W 4, -M 20) and the primers were removed by cutadapt (https://github.com/marcelm/cutadapt/). In average, 65866 and 73408 good-quality reads were retained for bacteria and fungi respectively. Using the QIIME 2 pipeline (2022.11), the paired-end reads were merged and the feature table was generated using the dada2 denoise-paired plugin with chimeric reads trimmed[42]. The q2-feature classifier plugin was used to assign taxonomy to sequences using the Silva 138 (the confidence threshold was a default value of 0.7). In total, 98805 and 20903 amplicon sequence variants were generated for bacteria and fungi respectively.

## High-throughput qPCR based chip

The absolute abundance of genes involved in C, N, P, and S cycling were measured using a high-throughput qPCR-based chip (QMEC) on SmartChip Real-Time PCR System (WaferGen Biosystems, Fremont, USA) following manual instructions[43]. The gene names and function annotations are presented in Table S5. The 16 S rRNA gene was used as the reference gene. In brief, the qPCR amplification procedure was an initial denaturation at 95 °C for 10 min, followed by 40 cycles of 30 s denaturation at 95 °C, 30 s annealing at 58 °C and 30 s extension at 72 °C. Each DNA sample was amplified in triplicate. If the results amplification efficiencies of <1.8 and > 2.2, negative control amplified and a threshold cycle (CT) > 31, they were excluded from further analyzes. In addition, bacterial and fungal absolute copy numbers were measured by qPCR using the same primers for 16 S and ITS sequencing, representing soil total bacterial and fungal abundance, respectively.

## Metagenomics

To examine the microbial taxa containing the specific genes involved in C, N, P, and S cycling, we sent the DNA samples for metagenomics sequencing. Paired-end sequencing was performed on the instrument NovaSeq 6000. The quality of raw reads with a sequencing depth of 20 G was controlled by using the Fastp (version 0.20.0)[44]. The filtered reads were assembled to contigs using the Megahit (version 1.1.2) via de Bruijn graph and with the minimum and maximum k-mer sizes of 21 and 121 respectively and step size of 10[45]. Contigs with the length > 500 bp were retained. Prodigal (version 2.6.3) was then used to identify the open reading frames in contigs and predict the protein-coding genes[46]. The KEGG functional annotations were conducted using Diamond (version 0.8.35) against the Kyoto Encyclopedia of Genes and Genomes database (version 94.2) with an e-value cutoff of ($<1 \times 10^{-5}$)[47]. To obtain the taxonomic assignments corresponding to specific functional genes, the sequences of predicted genes from the KEGG database were annotated based on the NCBI NR database using blastp in Diamond with e-value cutoff of $<1 \times 10^{-5}$.

## Soil incubation experiments

To verify the relationship between soil micronutrients and soil microbiome, we conducted a soil incubation experiment by adding Fe and Zn into five soils with contrasting chemical properties. The Fe and Zn were chosen for validation experiments because their concentrations correlated with most microbial parameters. Soils were collected from geographically distant sites across China, specifically Guangdong (GD, 110°6'6"E, 21°8'2"N), Jiangsu (JS, 118°38'4"E, 32°28'52"N), Shandong (SD, 116°19'50"E, 35°59'44"N), Yunnan (YN, 100°20'6"E, 25°34'47"N) and Neimenggu (NM, 117°44'18"E, 44°12'58"N) provinces.

The total Fe concentrations in the soils collected from GD, JS, SD, YN, and NM sites were 83231, 34381, 26920, 28500 and 10193 mg kg$^{-1}$ respectively. The total Zn concentrations were 73, 55, 82, 42, and 24 mg kg$^{-1}$ respectively. The concentrations of ferric ($FeCl_3$) and ferrous chlorides ($FeCl_2$) (1:1) and zinc chloride ($ZnCl_2$) dissolved in water were 1000 mg/kg and 100 mg/kg respectively, and the soils without micronutrient addition were the controls. The soil samples were incubated in plastic bags under darkness at 25 °C, while maintaining the moisture at 60% of the water-holding capacity. All the treatments had three replicates. After a two-week incubation, samples were collected for DNA extraction, 16 S sequencing, and gene abundance measurements. The methods employed for DNA extraction, sequencing, and qPCR were consistent with the procedures described above. The available Fe and Zn concentrations were measured according to the method described above[39]. While unspiked soils had Fe and Zn average concentrations of 8.7 and 1.2 mg/kg respectively at the end of the incubation, the Fe spiked soils had the Fe average concentration of 22.6 mg/kg and the Zn spiked soils had a concentration of 4.9 mg/kg (3 and 4 times greater availability in Fe or Zn spiked soils).

## Data analysis

Principal component analysis (PCA) in SPSS statistics software (version 24.0)[48] was used to group the total and available concentrations of Fe, Mn, Cu, Zn, Mo, and Ni. Correlations between these micronutrients are presented in Table S6. The principal component 1 (TPC1) and principal component 2 (TPC2) explained 51.4% and 19.5% of total micronutrients respectively (Fig. S10). The TPC1 mainly explained the variables of total Fe, Cu, Zn, and Ni, and the TPC2 explained the total Mn and Mo. Similarly, the principal component 1 (APC1) and principal component 2 (APC2) explained 43.1% and 25.8% of available micronutrients respectively (Fig. S10). The APC1 mainly explained the variables of available Fe, Cu, and Zn, and the APC2 explained the available Mn, Mo, and Ni.

Bacterial and fungal alpha diversity, i.e. Shannon and Chao 1 index, were obtained before rarefying all sequences at a minimum number of sequences per sample, 33018 and 35115, respectively[40]. Microbial beta diversity, i.e. PCoA1 and PCoA2, based on the weighted unifrac distance matrix was performed using the R package "phyloseq"[49]. The differences in the overall microbial community between different climate zones and land uses were tested by two-way ANOVA (Fig. S8). The differences in the overall microbial community between the soils with low, medium, and high concentrations of total and available micronutrients were investigated using PERMANOVA[50]. Given that the overall microbial community between soils with low and high micronutrient concentrations in most cases were significant (Tables S7 and S8), the "DESeq2" (version 1.40.2) was performed to identify the highly correlated bacterial and fungal taxa at the genus level (Figs. S6 and S7). Genera with a log2-fold change in relative abundance >1 and an adjusted $p < 0.05$ were selected as the genera that were highly correlated to micronutrients[51]. Then, the relationship between correlated taxa (identified by Deseq2 with the lowest adjusted p-value) and micronutrient concentrations were conducted by Spearman correlations. Due to the potential heterogeneity of the samples collected across China, we used data normalization of log10 transformation for micronutrients to mitigate the impact of sample heterogeneity on the relative abundances of correlated genera. Results showed similar correlations as compared to no data normalization (Tables S1 and S2).

Bacterial and fungal co-occurrence networks were structured separately based on the Spearman correlation matrix with a cut-off correlation coefficient determined by RMT theory in an automatic fashion[52] and the p-value was corrected by the FDR method[53]. The topological parameters of total degree, average degree and average path length that represented microbial network connectivity were

calculated[54]. A higher network total degree and average degree and a lower average path length represent higher network connectivity.

Given that soil micronutrient concentrations and microbial parameters were both driven by soil pH as reported previously[4,24] the partial correlations were conducted between micronutrients (individual elements and principal components) and microbial total abundance, diversity, relative abundance of taxa and network topological parameters, with soil pH effects removed by the SPSS. The p-values for all correlation tests were corrected with the adjusted p-value < 0.05 which was considered to be significant[55]. Partial correlation is a statistical method describing the relationship between two variables meanwhile taking away the effect of another variable on this relationship[56]. Thus, the soil pH was set up as the co-variance (the variable whose effect was taken away) in the partial correlation model in SPSS. The partial correlations were also conducted between micronutrients and the gene abundances for C, N, P, and S cycling.

Structural equation modeling (SEMs) was constructed to quantify the contributions of soil properties and other environmental factors to ecosystem production. We hypothesized that soil micronutrients were directly associated with ecosystem production (first path)[17], and the micronutrients also benefited ecosystem production by affecting soil microbiome (second path)[13]. The original paths of SEM related to our hypothesis are presented in Fig. S11. We used the bacterial/fungal total abundance measured by qPCR as the variable of "abundance" in the model. The bacterial/fungal PCoA1 was used as the variable of "diversity" in the model. The parameter of average degree extracting from bacterial/fungal co-occurrence networks was used as the variable of "network connectivity". The microbial function was divided into four groups of C, N, P, and S cycling. The principal component analysis (PCA) of the gene abundances from each group was conducted, and the first principal component (i.e. PC1) was used as the variance for "function". The variable of soil micronutrients was used based upon the PC1 from the PCA of total and available micronutrients. The variable of soil inorganic macronutrients was used based upon the PC1 from the PCA of soil total N and P contents. Similarly, the climatic parameter was created by using the PC1 from the PCA of mean annual precipitation, mean annual temperature, and sunshine duration. The spatial variance was created by using the PC1 from the PCA of the longitude and latitude (at the resolution of meters). We used the PC1 as variables because this component explained the majority of all the variables grouped. The NDVI (normalized difference vegetation index) at the sampling time point that reflects net ecosystem productivity was used as the variable of ecosystem productivity in the SEM model. The NDVI was collected using the Moderate Resolution Imaging Spectroradiometer (MODIS) aboard NASA's Terra satellites. The categorical variables of soil moisture (available water storage capacity) and texture in the model were collected using the HWSD (Harmonized World Soil Database v 1.2). The SEM was constructed using AMOS 24.0 (SPSS, Chicago, IL, USA) with maximum likelihood estimation to fit the covariance matrix in the model, based on the model criterion of Chi-square ($p > 0.05$) and root mean square error of approximation (RMSEA < 0.05)[57].

### Reporting summary

Further information on research design is available in the Nature Portfolio Reporting Summary linked to this article.

## Data availability

Raw sequencing data were deposited in the Sequence Read Archive (SRA) with the accession number: PRJNA924284 (16S for bacteria), PRJNA940307 (ITS for fungi) and PRJNA1035420 (metagenomics). The source data used in this study are available in the Figshare database at https://doi.org/10.6084/m9.figshare.24597126.v1. The database of silva used for taxonomy assignments are available online (https://www.arb-silva.de). The database of Kyoto Encyclopedia of Genes and Genomes (KEGG) for functional assignment is available online (https://www.genome.jp/kegg/).

## Code availability

The codes for sequencing analyzes are publicly available in the Figshare database at https://doi.org/10.6084/m9.figshare.24596892.v1.

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

## Acknowledgements

This study was supported by the National Science Foundation of China (41991334 to J.X. and Z.D., 41721001 to J.X.), the National Key Research and Development Program of China (2019YFC1803704 to Z.D.) and the Modern Agricultural Industry Technology System of China (CARS-01 to J.X.). M.D-B. acknowledges support from TED2021-130908B-C41/AEI/10.13039/501100011033/Unión Europea NextGenerationEU/PRTR and from the Spanish Ministry of Science and Innovation for the I + D + i project PID2020-115813RA-I00 funded by MCIN/AEI/10.13039/501100011033.

## Author contributions

J.X. and Z.D. developed the original idea, designed the study, and got the funding. Z.D., M.D.B., E.M.J., P.C.B., and J.X. wrote and revised the manuscript. X.G., J.M., D.H. and X.W. performed the experimental studies. J.Lin, J.Luo, Z.D., D.H., and R.Z. carried out the analysis.

## Competing interests

The authors declare no competing interests.
