## [Peer Review File · Nature Communications]

Metallic micronutrients are associated with the structure and function of the soil microbiomeREVIEWER COMMENTS

Reviewer #1 (Remarks to the Author):

The study by Dai et al is interesting and investigates the role of micronutrients in soil ecosystem function. It is well written with few typographical errors. The importance of micronutrients in soil biogeochemistry is understudied and hence this research is welcome. Comparing the relationships between microbial and fungal communities is particularly useful. However, I have some questions about the methodology used.

- 1) Although the authors mention the range of soil conditions analyses, we do not have the breakdown of samples per soil type/condition. E.g. how many soils were collected under crop, grass or forest cover? Could there be a bias in the sampling procedure, were the samples spatially clustered? Were specific soil types represented by multiple samples, and was this even across soil types, land uses and climate conditions?
- 2) What sample storage conditions were used, both for the DNA and soil analyses? Samples were taken over two years - is this temporal variation included in the statistical analyses?
- 3) Regarding soil sampling, how were samples taken at each site? The authors suggest that samples were pooled – how many? What was the soil sampling arrangement at each site? Were the soil samples sieved?
- 4) Micronutrients may co-vary with variables, in addition to pH. The authors mention that contrasting soil types were used, from high to low weathering. However, soil type, or texture, is not included in their modelling. The authors also included climate data, but this probably is not enough to take into account variation in soil moisture, as this will also be affected by soil texture and drainage.
- 5) The SEM models don't include carbon and nitrogen specifically, and they were grouped under the macronutrients category. Considering the importance of SOM for soil ecosystem function, shouldn't SOM be included as a separate variable in the modelling work? The authors have information on land use (crop, grass and forests), so why is this not included in the SEM work?
- 6) The authors appear not to have performed the routine statistical tests one would normally carry out before going to more complex statistical analyses. In other words, why didn't the authors determine whether differences in soil microbial communities between different soil types, land uses, climatic conditions etc were significant and whether there are

interactions between these variables in shaping microbial community structure? Related to this, Deseq2 was used to determine which microbial taxa were enriched in low vs high micronutrient soils. However, was the overall microbial community structure different between these two groups? Did the authors use permanova or a similar test to determine this? It is unwise to carry out Deseq analyses if the difference between the groups is not significant.

Other comments:

Lines 191-192: justification for rarefying prior to determining alpha diversity – at least a reference to justify.

Line 195: space missing between brackets.

Lines 210-213. The authors state that they removed pH effects, but not how this was done (unless I am missing something). Could the authors explain how pH effect was removed?

Line 230-231: what level of resolution was spatial data used? Meters, kilometres?

Line 358 “participate” and not participated.

Reviewer #2 (Remarks to the Author):

The study by Dai et al explores and demonstrates an important relationship between soil micronutrients and the aspects of microbial community composition and microbial functional genes across 180 sites. Moreover, the study investigates the hypothesis that soil micronutrients may indirectly affect plant productivity through their effect on soil microbes using structural equation models. In light of the recent findings demonstrating the potential role of micronutrients in ecosystem processes, the topic of the study is highly relevant and timely. It is particularly interesting that the authors analyze and contrast the potential role of micronutrients, macronutrients and pH as predictors of microbial alpha and beta diversity, microbial abundance, microbial functional genes and the complexity of microbial co-occurrence networks.

While the study has a lot of potential to increase our understanding of the predictors of microbial communities and function across soil types, I suggest that the authors consider the following major points which could help improve the manuscript:

1. The text is generally not difficult to follow but there are still a number of sentences that

are not entirely clear. I suggest that the authors revise the text and make sure that all sentences are unambiguous.

2. The method section lacks a lot of detail needed to understand/reproduce the study. Most notably, the construction of structural equation models was not fully described, the choice of paths was not justified, it was not clear what were the initial models, how were they simplified, which paths were significant and what happened to the non-significant ones, why were the three models created separately instead of one model including all the parameters, how much variation in productivity was explained by the models. The other issues are pointed out below.

3. Throughout the text, the terminology the authors use regarding the effect of micronutrients could be problematic. This is an observational study and the authors should be careful when stating that micronutrients (or any other variable) affect or contribute to structuring microbial communities. Rather, they are correlated, associated, etc.

4. While there are not many studies on this topic, this is not the first study that explores the relationship between micronutrients and microbial communities. Notably, the study by Peng et al (2022) explored a similar topic and reached some of the same conclusions. The current study provides some new insights but would benefit from the inclusion of the findings of that study. Moreover, the discussion provides little interpretation of the findings and does not include literature (mainly from agricultural studies) which explored the role of e.g. addition of specific micronutrients on soil bacteria or fungi.

Specific comments:

L75: Not clear what is meant by 'potential interaction' at this point.

L83: This sentence is not entirely clear. Should it be that: 'microbial abundance, diversity and complexity of microbial interactions are found to be positively correlated with ecosystem productivity'?

L102: I suggest using the term 'co-occurrence networks' instead of 'potential interaction' throughout. See another comment about that below.

L103-104: '...contribution of micronutrients to the microbiome'. I believe that a more accurate phrasing is needed. For example the 'relationship between micronutrients and microbial community composition and functioning'.

L106-107: This study cannot evaluate the role that micronutrients play in structuring

microbial communities but the amount of variation in microbial communities micronutrients can explain (i.e. the strength of correlation between the two).

L 108-109: This sentence may be redundant as it does not provide any new information.

L133:135: Does this method successfully extract micronutrients from soils with different pH?

L146-149: Could the authors provide references to the original description of primers?

L150: Could the authors include the details about PCR mix and PCR conditions as well as other important steps of library preparation?

L155: Could the authors explain how was quality filtering performed, what was the threshold used? How many good-quality sequences were retained?

L158: How many OTUs were obtained for bacteria and fungi?

L165: Could the authors provide the list of genes in a supplementary table?

L178: Which software was used for statistical analyses? Could the authors specify it and cite it?

L180: For this and other PCA analyses, were there any highly correlated variables included? It would be useful to include a table showing a correlation between different soil nutrients.

L192: What was the rarefaction depth? I suggest showing rarefaction curves in the supplement.

L193: Isn't it the case that the NMDS axes scores shouldn't be used individually for further analysis? Using e.g. PCoA axes score could be more appropriate.

L197: Could the authors briefly explain this process of determining the threshold of the correlation coefficient? Were the P values corrected using the FDR method?

L199: I suggest not using the term 'potential interaction'. These are co-occurrence networks. They could represent interactions to some extent but they could also be a result of shared environmental niches etc. This can be argued in the discussion section.

L202: Could the authors explain how was this done? Which basic soil properties?

L203: I suggest saying that those taxa were highly correlated and not responsive (this study was not designed to examine the latter).

L209: For all these correlation tests, was a correction of P-values for multiple testing applied?

L227: See the previous comment on using correlated variables for PCA. C and N, for example, tend to be highly correlated and if used in PCA together they could skew the

scores of PCA axes.

L230: Why use 2 PCA scores from 2 variables? Or only PC1 was used for all the groups of variables? Could the authors specify this?

L232: There is no explanation about how the paths were constructed (i.e. the direct and indirect paths of different variables). They have to be based on some biologically sound assumptions backed by literature. For instance, climate can have both a direct effect on productivity and possibly an indirect one through its influence on soil properties. I suggest that the authors describe the choice of paths in SEM and specify if the paths were significant (retained in the model) or not.

L251: Like stated previously, I believe that 'responsive' is not the accurate term.

L258: Not clear what 'unique' means in this context.

L296: Rather than 'co-affected' they were correlated.

L310: It was not previously explained that 'bacterial' 'fungal' and 'functional' models were constructed separately. I suggest clearly describing the SEM model construction. Moreover, it is not clear why these separate models were created instead of one model that includes both bacteria and fungi and the functional genes.

L333: I believe that Fig.1 shows the correlation between micronutrients and the relative abundance of genera.

Fig.1 There is very little interpretation of these results in the discussion.

Fig.4 It is not entirely clear what the colours of the lines in the SEMs represent. If red is a direct effect, why is the direct arrow from pH to production blue? Moreover, could the authors explain how they produced the 'microbial function' variable?

Reference: Peng, Z., Liang, C., Gao, M. et al. The neglected role of micronutrients in predicting soil microbial structure. *npj Biofilms Microbiomes* 8, 103 (2022).

<https://doi.org/10.1038/s41522-022-00363-3>

Reviewer #3 (Remarks to the Author):

This manuscript conducted a regional survey of 180 locations in China, covering a wide range of soil conditions and investigating the relationship between metal micronutrients

and the abundance, diversity, and function of soil microbiota. This is a very time-consuming task. It did provide new insights into the relationship between metal micronutrients and soil microbiota. However, there are still some shortcomings. I attempt to provide suggestions in the comments below.

General comments:

1. The concept of exploring how the micronutrients affect the structure and function of soil microbiomes in terrestrial ecosystems by a regional scale survey is interesting and the author obtained some meaningful results. But I feel a bit worrying that the results from regional scale survey are not enough to support the results. As mentioned in the authors' conclusion, this conclusion can be better reflected if the results of some culture/incubation experiments are supplemented to validate.
2. Please improve the information of samples, such as what is the division of soil type? Please indicate whether the soil is rhizosphere soil or bulk soil. It is best to provide a distribution map of sampling points and annotate the sampling information to let people know the difficulty of this task.
3. Do you consider the impact of a single soil sample (with no duplicates at each sample point) on the results and what method did you adopt to solve it? Does your sample cover all of China? If the coverage is large, the heterogeneity of the sample will also be very different. It means you need to consider the impact of other factors in different locations on the relative abundance of species. This is because I observed that you used the concentration gradient of elements as a reference system (x-axis) in the results, which is different from usual laboratory concentration gradient experiments and more like a meta-analysis of a small sample. Generally, special data normalization methods need to be adopted to reduce the impact of sample heterogeneity. How did you evaluate and solve this problem?
4. How to determine the relationship between genes and trace elements? If using correlation, please indicate whether it is positive or negative and the p-value. In addition, you can also perform statistical analysis on positive and negative correlation results, which will allow us to understand the direction of gene regulation by elements, which is also very interesting.
5. It had to be said that the design of figure 3 is innovative, but please pay attention to the use of color schemes so that readers can identify it more clearly. And please increase the font size. Also, it is important to explain clearly how you distinguish between total and

available. Otherwise, I believe that the significance reflected by the total amount of micronutrients is not significant, which will greatly compromise the novelty of your article.

6. Can you clarify which microorganisms provide these genes? This is also crucial for functional traceability, as it will support the results of species being related to elements and elements being related to genes. But I did not find this result in your manuscript. You can choose to use metagenomics or other powerful methods to supplement this result.

7. What does the variable 'interaction' represent in the SEM model? How was it calculated, please specify in the material method. Also, are each path significant? Please indicate the p-value. Could quantitative data on functional genes be added to this SEM model, which will explain the contribution of species function to ecological productivity from a functional perspective. It will improve the completeness of your results.

8. It would be great to continue to pay attention to ecosystem production, but based on the main body of the manuscript, I think you may need to discuss the phenomenon that micronutrients do not affect ecosystem production by altering microbial function, which would be very interesting. At the same time, it is necessary to strengthen the logic of the discussion and pay attention to the progressive relationship of each paragraph. This should be gradual and in-depth, rather than flat and straightforward.

Specific Comments:

1. L94, "under changing conditions", Could the author specify what the "changing conditions" represent?
2. L113-114, Could the author provide the sampling map of the sites of the soil sample collection?
3. L148, Can the authors explain why only the V4 region was measured in 16S sequencing?
4. L290-291, Incomprehensible sentences.
5. L297, ..., soil pH and soil micronutrients (Fig. S9), I don't think most genes were co-affected by soil macronutrients in Fig S9, but only TMn.
6. L304-305, there was no available Fe in phosphorus cycling in Fig. 3e, please check it.
7. Please increase the font size of Figure 1 and distinguish between fungi and bacteria. What are the criteria for selecting species? Is it a significantly correlated species or a high abundance species? In figure 1, please pay attention to labeling p-value and interpretation rate to indicate the reliability of linear correlation.

8. Fig. 2 and Fig. S9, what the unique and shared OTUs meaning, author should provided some specific explanation in the Figure tile.
9. I do not understand the emphasis on the role of fusarium in the results, and there was no explanation given for this description in the discussion of the article. I am not sure if this is meaningful.
10. What is the meaning of 'mean' in Figures 3a and 3b? What does C: N, C: P, and N: P in Figure 3b mean? Please explain it. I don't think "Total+available" is make sense here, whether the author can explain or consider deleting it.
11. Figure 4c, pay attention to layout, I think there are some beyond the boundaries.
12. Please enlarge the font of figure 1, figure 2b and figure 2d. Please check all the figures and enhance the readability of them.
13. The placement and stitching of the figures in the supporting information are not standardized, and the author should use regular software to readjust them.
14. Fig. S1, the Y-axis title of pH plot should also be added. Additionally, (a)-(f) should also be added in Fig. S1, both (a)-(d) should also be added in Fig. S2.
15. Fig. S6, The figure lacks explanatory notes for the blue and red dots, please check it.
16. Please indicate the number of upward and downward adjustments in figure S7.
17. Some of the results in the manuscript lack significant annotations (p-value), and I have already raised some of them. Please review the entire manuscript as this will affect the reliability of your results.

Reviewer #1 (Remarks to the Author):

The study by Dai et al is interesting and investigates the role of micronutrients in soil ecosystem function. It is well written with few typographical errors. The importance of micronutrients in soil biogeochemistry is understudied and hence this research is welcome. Comparing the relationships between microbial and fungal communities is particularly useful. However, I have some questions about the methodology used.

1) Although the authors mention the range of soil conditions analyses, we do not have the breakdown of samples per soil type/condition. E.g. how many soils were collected under crop, grass or forest cover? Could there be a bias in the sampling procedure, were the samples spatially clustered? Were specific soil types represented by multiple samples, and was this even across soil types, land uses and climate conditions?

1. Thank you for your positive and constructive comments. We aimed to address all of them below. To clearly present the soil conditions, we have provided a distribution map of sampling points and annotate sampling information. Please see Fig. S1, Line 120-121 and Line 122-124.

2) What sample storage conditions were used, both for the DNA and soil analyses? Samples were taken over two years - is this temporal variation included in the statistical analyses?

2. We apologies for the lack of clarity in the previous version of our manuscript wherein soil sampling and storage information were not properly described. Briefly, we stored air-dried soil samples in a refrigerator at 4 °C, under which the chemical properties are not expected to change over years. We stored the soil samples for DNA extraction in -80 °C, under which the soil microbiome is not expected to change. We conducted simultaneous analyses of both the microbiome and soil properties to prevent the introduction of systematic errors. In the revised version of our manuscript, we have included additional information regarding the criteria for soil sample collection and storage. Please see Line 122-129.

3) Regarding soil sampling, how were samples taken at each site? The authors suggest that samples were pooled – how many? What was the soil sampling arrangement at each site? Were the soil samples sieved?

3. Please see the response #2 to this reviewer. This information has been provided in Line 122-125.

4) Micronutrients may co-vary with variables, in addition to pH. The authors mention that contrasting soil types were used, from high to low weathering. However, soil type, or texture, is not included in their modelling. The authors also included climate data, but this probably is not enough to take into account variation in soil moisture, as this will also be affected by soil texture and drainage.

4. Thank you for the constructive comment. We have added soil texture in the SEM modelling. As we know, soil types are representatively reflected by their physical,

chemical and biochemical properties such as soil pH, organic C, nitrogen, texture, etc. Therefore, we decided to include soil texture (instead of soil type) as a variable.

In general, a large scale moisture is associated with aridity or precipitation. Drier ecosystems are more likely to be dry. However, the climate data, especially the annual mean precipitation only partially explain the variation of soil moisture, as moisture is also affected by drainage. As requested by reviewer, we have also added the parameter of moisture in the SEM model. Please see the revised Fig. 5.

To avoid the misunderstanding, we use the “soil conditions” to replace the “soil type” in the Method section. Please see Line 117.

5) The SEM models don't include carbon and nitrogen specifically, and they were grouped under the macronutrients category. Considering the importance of SOM for soil ecosystem function, shouldn't SOM be included as a separate variable in the modelling work? The authors have information on land use (crop, grass and forests), so why is this not included in the SEM work?

5. We agree that the carbon and nitrogen in the SEM model should be separated variables. Thus, we have re-run the SEM model by using the total C as a separated variable. Please see the revised Fig. 5.

Yes, it is a good try to include land use in the SEM work. Actually, we tried to put the land use in the SEM model as requested by the reviewer, while the model did not fit very well and did not meet the significance criteria. Considering different land uses resulted in different soil properties, i.e. one land use type represents a unique soil physical and chemical property such as pH, C, N, etc, thus the land use variance could be replaced by soil physiochemical properties in the model. In addition, adding one more variance would introduce many paths into the model and make the model more complex and unreadable. Finally, we decided not to include land use as a variable in SEM model and use soil basic properties in the model, which was more associated with our hypothesis that soil physical and chemical properties (including both macronutrient and micronutrient) correlated with the structure and function of microbiome and further the ecosystem production.

6) The authors appear not to have performed the routine statistical tests one would normally carry out before going to more complex statistical analyses. In other words, why didn't the authors determine whether differences in soil microbial communities between different soil types, land uses, climatic conditions etc were significant and whether there are interactions between these variables in shaping microbial community structure? Related to this, Deseq2 was used to determine which microbial taxa were enriched in low vs high micronutrient soils. However, was the overall microbial community structure different between these two groups? Did the authors use permanova or a similar test to determine this? It is unwise to carry out Deseq analyses if the difference between the groups is not significant.

6. Thank you for this good comment. We did not perform the routine statistical test as our main goal is to investigate the relationship between micronutrients and soil microbial communities. But knowing the microbial communities between soil types,

land uses, climatic conditions etc are also important, though they have been reported by previous papers. Thus, we have analysed the differences in microbial communities between land uses and climatic conditions, and also their interactions. Please see the Fig. S11 and Line 435-438.

Related to Deseq2, we have investigated the differences in the overall microbial community structure between low and high micronutrient soils. Please see Table S3, Table S4 and Line 263-265. We have found that the differences between low and high concentration soils for most micronutrients were significant. So it was necessary to carry out Deseq analyses. Although some differences were not significant, we still think it is necessary to carry out Deseq, because there may be some differences in the abundances of specific genera, although the overall community composition was not significant.

Other comments:

Lines 191-192: justification for rarefying prior to determining alpha diversity – at least a reference to justify.

Authors: Please see the revision in Line 251-252.

Line 195: space missing between brackets.

Authors: Please see the correction in Line 254.

Lines 210-213. The authors state that they removed pH effects, but not how this was done (unless I am missing something). Could the authors explain how pH effect was removed?

Authors: The details in how to run partial correlation have been described in Line 281-284.

Line 230-231: what level of resolution was spatial data used? Meters, kilometres?

Authors: The values of longitude and latitude at highest resolution were recorded when we were collecting soil samples at each site. The level of resolution is meters. Please see Line 309.

Line 358 “participate” and not participated.

Authors: The sentence has been deleted during the revision.

Reviewer #2 (Remarks to the Author):

The study by Dai et al explores and demonstrates an important relationship between soil micronutrients and the aspects of microbial community composition and microbial functional genes across 180 sites. Moreover, the study investigates the hypothesis that soil micronutrients may indirectly affect plant productivity through their effect on soil microbes using structural equation models. In light of the recent findings demonstrating the potential role of micronutrients in ecosystem processes, the topic of the study is highly relevant and timely. It is particularly interesting that the authors analyze and

contrast the potential role of micronutrients, macronutrients and pH as predictors of microbial alpha and beta diversity, microbial abundance, microbial functional genes and the complexity of microbial co-occurrence networks.

While the study has a lot of potential to increase our understanding of the predictors of microbial communities and function across soil types, I suggest that the authors consider the following major points which could help improve the manuscript:

1) The text is generally not difficult to follow but there are still a number of sentences that are not entirely clear. I suggest that the authors revise the text and make sure that all sentences are unambiguous.

I. The text and sentences of the manuscript have been double-checked and the language has been edited by our native English speaker.

2) The method section lacks a lot of detail needed to understand/reproduce the study. Most notably, the construction of structural equation models was not fully described, the choice of paths was not justified, it was not clear what were the initial models, how were they simplified, which paths were significant and what happened to the non-significant ones, why were the three models created separately instead of one model including all the parameters, how much variation in productivity was explained by the models. The other issues are pointed out below.

2. Thank you for the comment. The line arrows in the previous SEM were significant paths. To clarify the initial models, we have added the original paths of SEM in Fig. S5 and Line 293-296. In the original paths, we hypothesize that soil micronutrients and other properties would affect ecosystem production by two paths: 1) these parameters directly affected ecosystem production, we call “direct effects”; 2) the parameters affected soil microbiome, and the changes in soil microbiome further affect the ecosystem production, we call “indirect effects”.

We only showed the arrows which represented the significant effects in the SEM model, and the non-significant ones were not presented. As described above, putting non-significant arrows in the model made the figure more complex and the readers may not get the key point immediately. These explanations have been added in the Figure caption, Fig. 5.

We have compared the models including three models together and separately, and finally decided to use three models separately, due to: 1) the presentation of data. Many variables together showed a very complicated model (if we put all the variables together there should be more than 50 paths), which presenting a messy vision for readers. So the presentation of SEM separately would be better to read; 2) the logic of paths. Corresponds to Fig. 2 and Fig. 3, we discussed the effects of micro-nutrients on bacteria, fungi and function separately. So creating models separately can give a direct and clear logic for readers to understand how micronutrients affected ecosystem production via affecting bacteria, fungi and their function.

In addition, the explained variation of productivity have been added. Please see revised Fig. 5.

3. Throughout the text, the terminology the authors use regarding the effect of micronutrients could be problematic. This is an observational study and the authors should be careful when stating that micronutrients (or any other variable) affect or contribute to structuring microbial communities. Rather, they are correlated, associated, etc.

3. Thanks. We have changed some terminology associated with observational study using the more precise terms such as correlated, associated, etc. Please see the revisions marked with yellow colors throughout the manuscript.

As required by reviewer 3, we have conducted an incubation test to valid the findings. So related to the results from incubation experiments, we have retained some terminologies regarding “the effect of micronutrients”.

4. While there are not many studies on this topic, this is not the first study that explores the relationship between micronutrients and microbial communities. Notably, the study by Peng et al (2022) explored a similar topic and reached some of the same conclusions. The current study provides some new insights but would benefit from the inclusion of the findings of that study. Moreover, the discussion provides little interpretation of the findings and does not include literature (mainly from agricultural studies) which explored the role of e.g. addition of specific micronutrients on soil bacteria or fungi.

4. The study by Peng et al (2022) actually explored a similar topic and reached some of the same conclusions. Actually we initiated the idea and started the work in 2018. In our study, we removed the pH effects which may interfere the conclusion from the correlation analysis. Notably, we have conducted the relationship between micronutrients and soil function involved in C, N, P and S cycles, and the relationship between soil micronutrients and ecosystem production. These are the novelties in our work, which have not been reported previously. Requested by reviewer, we fully appreciate the study by Peng et al (2022) and give some discussions of our findings to that literature. Please see the Line 442-446, Line 450-454 and Line 465-466 in the discussion.

Specific comments:

L75: Not clear what is meant by ‘potential interaction’ at this point.

Authors: We have changed the “potential interaction” to “connection”. Please see Line 77.

L83: This sentence is not entirely clear. Should it be that: ‘microbial abundance, diversity and complexity of microbial interactions are found to be positively correlated with ecosystem productivity’?

Authors: This sentence has been revised. Please see Line 85-86.

L102: I suggest using the term ‘co-occurrence networks’ instead of ‘potential interaction’ throughout. See another comment about that below.

Authors: Thanks. We have changed to the “potential interaction” in the whole manuscript to “connection” or “co-occurrence networks”.

L103-104: ‘...contribution of micronutrients to the microbiome’. I believe that a more accurate phrasing is needed. For example the ‘relationship between micronutrients and microbial community composition and functioning’.

Authors: Thanks. Please see the revision in Line 106-107.

L106-107: This study cannot evaluate the role that micronutrients play in structuring microbial communities but the amount of variation in microbial communities micronutrients can explain (i.e. the strength of correlation between the two).

Authors: Thanks. Please see the revision in Line 109-110.

L 108-109: This sentence may be redundant as it does not provide any new information.

Authors: This sentence has been deleted. Please see Line 111.

L133:135: Does this method successfully extract micronutrients from soils with different pH?

Authors: Yes. This method was extensively used for most soil types, and was an appropriate technique for large-scale observation study. Please see the explanation in Line 145.

L146-149: Could the authors provide references to the original description of primers?

Authors: The references have been provided. Please see Line 157 and Line 160.

L150: Could the authors include the details about PCR mix and PCR conditions as well as other important steps of library preparation?

Authors: Please see the details about PCR mix and PCR conditions in Line 160-167.

L155: Could the authors explain how was quality filtering performed, what was the threshold used? How many good-quality sequences were retained?

Authors: Thank you for pointing this. After we double-checked the pipeline, we actually used Fastp and QIIME 2 to process the data. So the description of this pipeline has been corrected. Please see Line 173-176. The details for quality filtering and the effective of good-quality sequences retained have been added. Please see Line 169-172.

L158: How many OTUs were obtained for bacteria and fungi?

Authors: Please see Line 177-178.

L165: Could the authors provide the list of genes in a supplementary table?

Authors: The genes have been added as Table S1. Please see Table S1 and Line 186.

L178: Which software was used for statistical analyses? Could the authors specify it and cite it?

Authors: The software and citation have been input. Please see Line 238-240.

L180: For this and other PCA analyses, were there any highly correlated variables included? It would be useful to include a table showing a correlation between different soil nutrients.

Authors: We used principal component analysis, because it transforms a set of potentially correlated variables into a set of linearly uncorrelated variables. We have added the results of correlations between different soil nutrients, while the variables were not highly correlated. Please see Table S2 and Line 240.

L192: What was the rarefaction depth? I suggest showing rarefaction curves in the supplement.

Authors: The rarefaction curves for all 180 samples can not be clearly presented in one figure. Instead, we use the rarefaction depth to replace the curves. Please see the Line 251-252.

L193: Isn't it the case that the NMDS axes scores shouldn't be used individually for further analysis? Using e.g. PCoA axes score could be more appropriate.

Authors: Thank for the suggestion. We used the PCoA axes score instead of NMDS for all the analyses in our study. Please see Line 252-253. Correspondingly, the Fig. 2 and Fig. 5 have been revised by using PCoA scores.

L197: Could the authors briefly explain this process of determining the threshold of the correlation coefficient? Were the P values corrected using the FDR method?

Authors: Please see the revision in Line 254-257.

L199: I suggest not using the term 'potential interaction'. These are co-occurrence networks. They could represent interactions to some extent but they could also be a result of shared environmental niches etc. This can be argued in the discussion section.

Authors: All the “potential interaction” have been changed to “connection” or “co-occurrence networks” throughout the manuscript.

L202: Could the authors explain how was this done? Which basic soil properties?

Authors: Considering this figure did not contribute greatly to the interpretation of major finding. Also, the manuscript was limited by length. Thus, we have deleted this figure.

L203: I suggest saying that those taxa were highly correlated and not responsive (this study was not designed to examine the latter).

Authors: Please see the revision in the Line 262, 267, and others with yellow color marked throughout the manuscript.

L209: For all these correlation tests, was a correction of P-values for multiple testing applied?

Authors: Yes, the p-values for all the correlation tests have been corrected. Please see

Line 286-288.

L227: See the previous comment on using correlated variables for PCA. C and N, for example, tend to be highly correlated and if used in PCA together they could skew the scores of PCA axes.

Authors: The correlation between soil macronutrient have been conducted. None of them had a high correlation. Please see the Table S2.

L230: Why use 2 PCA scores from 2 variables? Or only PC1 was used for all the groups of variables? Could the authors specify this?

Authors: Please see the explanation in Line 309-311.

L232: There is no explanation about how the paths were constructed (i.e. the direct and indirect paths of different variables). They have to be based on some biologically sound assumptions backed by literature. For instance, climate can have both a direct effect on productivity and possibly an indirect one through its influence on soil properties. I suggest that the authors describe the choice of paths in SEM and specify if the paths were significant (retained in the model) or not.

Authors: Please see the response #2 to this reviewer.

L251: Like stated previously, I believe that 'responsive' is not the accurate term.

Authors: The "responsive" has been changed to "correlated". Please see Line 335 and others throughout the manuscript.

L258: Not clear what 'unique' means in this context.

Authors: This sentence has been already deleted due to the previous Fig. 2b and d deletion.

L296: Rather than 'co-affected' they were correlated.

Authors: This sentence has been already deleted due to the previous Fig. 2b, Fig. 2d and Fig. S9 deletion.

L310: It was not previously explained that 'bacterial' 'fungal' and 'functional' models were constructed separately. I suggest clearly describing the SEM model construction. Moreover, it is not clear why these separate models were created instead of one model that includes both bacteria and fungi and the functional genes.

Authors: Please see the response#2 to this reviewer.

L333: I believe that Fig.1 shows the correlation between micronutrients and the relative abundance of genera.

Authors: This sentence has been deleted during the revision.

Fig.1 There is very little interpretation of these results in the discussion.

Authors: The interpretation of these results have been discussed in the discussion.

Please see Line 500-507.

Fig.4 It is not entirely clear what the colours of the lines in the SEMs represent. If red is a direct effect, why is the direct arrow from pH to production blue? Moreover, could the authors explain how they produced the 'microbial function' variable?

Authors: After the revision, the red and blue lines represented the significantly negative and positive correlations. Please see the response#2 to this reviewer.

Reference: Peng, Z., Liang, C., Gao, M. et al. The neglected role of micronutrients in predicting soil microbial structure. *npj Biofilms Microbiomes* 8, 103 (2022). <https://doi.org/10.1038/s41522-022-00363-3>

Reviewer #3 (Remarks to the Author):

This manuscript conducted a regional survey of 180 locations in China, covering a wide range of soil conditions and investigating the relationship between metal micronutrients and the abundance, diversity, and function of soil microbiota. This is a very time-consuming task. It did provide new insights into the relationship between metal micronutrients and soil microbiota. However, there are still some shortcomings. I attempt to provide suggestions in the comments below.

General comments:

1) The concept of exploring how the micronutrients affect the structure and function of soil microbiomes in terrestrial ecosystems by a regional scale survey is interesting and the author obtained some meaningful results. But I feel a bit worrying that the results from regional scale survey are not enough to support the results. As mentioned in the authors' conclusion, this conclusion can be better reflected if the results of some culture/incubation experiments are supplemented to validate.

1. We have conducted an incubation experiment to validate some finding of observation study. We mainly focus on how the Fe and Zn, the highly correlated parameters, affected the bacterial diversity, community and the functional gene abundances. Please see the Line 211-234 in the Method section, Line 393-411 in the Result section and Line 435-438 and Line 491-499 in the Discussion section.

2. Please improve the information of samples, such as what is the division of soil type? Please indicate whether the soil is rhizosphere soil or bulk soil. It is best to provide a distribution map of sampling points and annotate the sampling information to let people know the difficulty of this task.

2. The sample information and distribution map have been provided. Please see Fig. S1. The details in soil sample criteria have been added in the manuscript. Please see Line 122-124.

3) Do you consider the impact of a single soil sample (with no duplicates at each sample point) on the results and what method did you adopt to solve it? Does your

sample cover all of China? If the coverage is large, the heterogeneity of the sample will also be very different. It means you need to consider the impact of other factors in different locations on the relative abundance of species. This is because I observed that you used the concentration gradient of elements as a reference system (x-axis) in the results, which is different from usual laboratory concentration gradient experiments and more like a meta-analysis of a small sample. Generally, special data normalization methods need to be adopted to reduce the impact of sample heterogeneity. How did you evaluate and solve this problem?

3. The distribution of soil samples have been added in Fig. S1. To reduce the heterogeneity of the sample raised by the reviewer, we have used the data normalization method, i.e. $\text{Log}_{10}X$, to normalize sample heterogeneity. The correlation between the relative abundances of genera and normalized micronutrients are presented in Table S5 and Table S6. The results from data normalization was mostly same as the finding by using concentration gradient. Please see the Table S5 and Table S6 and description in the Line 269-273 in the Methods.

4) How to determine the relationship between genes and trace elements? If using correlation, please indicate whether it is positive or negative and the p-value. In addition, you can also perform statistical analysis on positive and negative correlation results, which will allow us to understand the direction of gene regulation by elements, which is also very interesting.

4. Actually, we used the partial correlation (with pH effect removed) to investigate the relationship between micronutrients and specific gene abundances (Please see Line 281-284). The positive and negative correlation coefficients and the p-value have been added in Table S7. Also, the Fig. 3 have been revised to clearly show the positive and negative correlations. Please see Fig. 3. As soils samples were not over-contaminated, most of the correlations of the genes presented in the C, N, P and S pathways are positive (We found some genes negatively correlated with micronutrients, while they are not significant).

5) It had to be said that the design of figure 3 is innovative, but please pay attention to the use of color schemes so that readers can identify it more clearly. And please increase the font size. Also, it is important to explain clearly how you distinguish between total and available. Otherwise, I believe that the significance reflected by the total amount of micronutrients is not significant, which will greatly compromise the novelty of your article.

5. The figure 3 has been revised according to the reviewer's comments by using proper color schemes, increasing font size and distinguish between total and available micronutrients. Please see Fig. 3.

6) Can you clarify which microorganisms provide these genes? This is also crucial for functional traceability, as it will support the results of species being related to elements and elements being related to genes. But I did not find this result in your

manuscript. You can choose to use metagenomics or other powerful methods to supplement this result.

6. Requested by the reviewer, we have sent the DNA samples for metagenomics, and used bioinformatics analysis to find the microorganisms that contain specific genes. The processing methods are presented in Line 196-209 in the Method section. The microorganism type have been added in Table S8. We also discussed this part in the Line 388-391 in the Results and Line 507-514 in the Discussion section.

7) What does the variable 'interaction' represent in the SEM model? How was it calculated, please specify in the material method. Also, are each path significant? Please indicate the p-value. Could quantitative data on functional genes be added to this SEM model, which will explain the contribution of species function to ecological productivity from a functional perspective. It will improve the completeness of your results.

7. Actually, the “interaction” in the SEM model represented the taxa connections of the co-occurrence networks. To avoid the ambiguity, we have replaced all the “interaction” by “connection” throughout the manuscript. In the model, we used the parameter of “average degree” extracted from occurrence network. Please see the explanation in the Line 299-300.

We only presented the significant path in the SEM model, please see the reasons in the response#2 to reviewer 2.

Actually, the SEM model showed the contribution of microbial function to ecological productivity from a functional perspective, while the methods have not been clearly described. Therefore, we have added the details in the Method section. Please see Line 291-319. In addition, we have improved the microbial function SEM model by dividing the quantitative data functional genes into four groups, i.e. C, N, P and S cycling. Please see new Fig. 5c. Please also see the Line 300-303 in the Method section, Line 422-427 in Result section and Line 552-559 in the Discussion.

8) It would be great to continue to pay attention to ecosystem production, but based on the main body of the manuscript, I think you may need to discuss the phenomenon that micronutrients do not affect ecosystem production by altering microbial function, which would be very interesting. At the same time, it is necessary to strengthen the logic of the discussion and pay attention to the progressive relationship of each paragraph. This should be gradual and in-depth, rather than flat and straightforward.

8. The function SEM model have been revised. Please see the response#7 to this reviewer. Thus, the discussion have been also revised. Please see the Line 553-559.

Furthermore, the writing of discussion have been considerable revised. Please see the yellow color marked lines in the Discussion section.

Specific Comments:

1. L94, “under changing conditions”, Could the author specify what the “changing conditions” represent?

Authors: Sorry for the ambiguity. We have deleted this phrase. Please see Line 96.

2. L113-114, Could the author provide the sampling map of the sites of the soil sample collection?

Authors: The sampling map has been added as Fig. S1.

3. L148, Can the authors explain why only the V4 region was measured in 16S sequencing?

Authors: The reference has been added and the primers target for V4 region have been widely used and published. Please see Line 157.

4. L290-291, Incomprehensible sentences.

Authors: Please see the revision in Line 370.

5. L297, ..., soil pH and soil micronutrients (Fig. S9), I don't think most genes were co-affected by soil macronutrients in Fig S9, but only TMn.

Authors: This sentence has been already deleted due to the previous Fig. S9 deletion.

6. L304-305, there was no available Fe in phosphorus cycling in Fig. 3e, please check it.

Authors: The available Fe had no effects on P cycling. Please see the revision in Line 386.

7. Please increase the font size of Figure 1 and distinguish between fungi and bacteria. What are the criteria for selecting species? Is it a significantly correlated species or a high abundance species? In figure 1, please pay attention to labeling p-value and interpretation rate to indicate the reliability of linear correlation.

Authors: The font size has been increased, and the bacteria and fungi have been distinguished by labelling 'B' and 'F'. Please see the revised Fig. 1.

The p-value and interpretation rate have also been added. Please see the Table S5 and Table S6.

The genera used in Fig. 1 were identified as the significantly correlated genera with micronutrients by Deseq2 with the lowest adjusted-p value (i.e. most significant). Please see the caption in Fig. 1.

8. Fig. 2 and Fig. S9, what the unique and shared OTUs meaning, author should provided some specific explanation in the Figure tile.

Authors: These figures have been deleted.

9. I do not understand the emphasis on the role of fusarium in the results, and there was no explanation given for this description in the discussion of the article. I am not sure if this is meaningful.

Authors: We mention the fusarium in the results because we want to show some fungal genera can be potentially bio-indicators for Mo and Ni in soils. Thus, we have added a

discussion about it. Please see Line 501-502.

10. What is the meaning of 'mean' in Figures 3a and 3b? What does C: N, C: P, and N: P in Figure 3b mean? Please explain it. I don't think "Total+available" is make sense here, whether the author can explain or consider deleting it.

Authors: The "mean" represented the average number of genes correlated with all micronutrients or soil macronutrient+pH. Please see the explanation in the caption in Fig. 3. The "Total+available" bars have been deleted as suggested by reviewers.

11. Figure 4c, pay attention to layout, I think there are some beyond the boundaries.

Authors: This has been revised.

12. Please enlarge the font of figure 1, figure 2b and figure 2d. Please check all the figures and enhance the readability of them.

Authors: The font of these figures have been enlarged. Please see Fig.1. The Fig. 2b and Fig. 2d have been deleted.

13. The placement and stitching of the figures in the supporting information are not standardized, and the author should use regular software to readjust them.

Authors: The figures in the supporting information have been adjusted.

14. Fig. S1, the Y-axis title of pH plot should also be added. Additionally, (a)-(f) should also be added in Fig. S1, both (a)-(d) should also be added in Fig. S2.

Authors: The Y axis of pH plot has been added. Please see the new Fig. S2.

The (a)-(f) and (a)-(d) have been added in new Fig. S2 and Fig. S3 respectively.

15. Fig. S6, The figure lacks explanatory notes for the blue and red dots, please check it.

Authors: The explanatory notes for the blue and red dots have been added. Please see the captions in Fig. S8 and Fig. S9.

16. Please indicate the number of upward and downward adjustments in figure S7.

Authors: The number of upward and downward adjustments have been added in the new Fig. S8.

17. Some of the results in the manuscript lack significant annotations (p-value), and I have already raised some of them. Please review the entire manuscript as this will affect the reliability of your results.

Authors: We have checked from the manuscript and make sure all the (p-value) have been added. Most information has been added in Supporting Information. Please see Table S4-7.

REVIEWER COMMENTS

Reviewer #1 (Remarks to the Author):

I read the response to my comments which were for the most part addressed. I also note the considerable revisions done in response to the other reviewers. There are some remaining questions.

Regarding point 3 in the responses to my comments, can the authors clarify what they mean by a "site"? Is it an area of a specific size, such as 10 x 10 m or something similar? Were there any criteria for choosing sites (such as avoiding certain features)? If this was described elsewhere, can this be shown in section 2.1?

Regarding point 4: I note that the authors included soil moisture in the SEM analysis, but how was soil moisture quantified?

Regarding the point about rarefaction and microbial alpha diversity (lines 251 and 252 of the revised manuscript): To justify their approach, the authors use a reference from 2011 which is key to the method, but the discussion about rarefying or not prior to alpha diversity analyses developed significantly since then. To my understanding, it is still not advised to rarefy prior to alpha diversity calculation. Some references discussing this include [10.3389/fmicb.2019.02407](https://doi.org/10.3389/fmicb.2019.02407) and [10.1016/j.csbj.2021.12.036](https://doi.org/10.1016/j.csbj.2021.12.036). If the authors can justify their approach with a recent and relevant reference, I am happy with it, otherwise, I would advise re-calculation of alpha diversity using an approach that reduces biases associated with rarefaction, for example by pooling samples as done in [10.1016/j.csbj.2021.12.036](https://doi.org/10.1016/j.csbj.2021.12.036).

Reviewer #2 (Remarks to the Author):

The authors performed a thorough revision of the manuscript with detailed responses to all the major concerns and they included additional data and analyses. Particular compliments to the authors for conducting a small incubation study that evaluated the influence of Fe and Zn additions on microbial community composition and functional genes.

Below are several additional points that the authors could take into account to clarify some remaining issues.

L39-40: It is unclear what happens with increasing micronutrients. The incubation experiment could receive a clearer introduction. For instance, the authors could state that they conducted an experiment with Zn and Fe additions for five different soil types showing that these micronutrients can indeed affect microbial community composition and functioning.

L92: are associated with productivity

L96: to predict ecosystem responses to...? The sentence is not finished.

L104: It is still not entirely clear here (and elsewhere) what the term 'connection' refers to without adding the context. It could be 'fungal and bacterial network connectivity' and later, for short just 'network connectivity'.

L 109-110: Perhaps a clearer version of this sentence would be something like: '... that micronutrients can explain a unique portion of the variation in soil microbiome structure and function in addition to pH and macronutrients'. In any case: the word 'except' here suggests that pH and macronutrients were not considered.

L 140: Could the authors add a reference to the method for the analyses of total micronutrients?

L 197: 'to find' or perhaps better 'examine'

L 254: Could the authors clarify here that bacterial and fungal co-occurrence networks were constructed separately? Otherwise, it could be interpreted as a mixed microbial network.

L 260: This is not entirely clear. It could be stated that a higher network degree and average path length represent higher network connectivity.

L 261-269: This part requires some attention. It is not evident to what these taxa are correlated (L 262) (to each other, to micronutrients,... or were they 'indicators' of soils with low or high micronutrient concentrations?). What was the criterion for dividing the soils into these two groups and which test was used to determine the difference in microbial community composition? I suggest that the authors consider the following order of description of the steps: 1. They divided the soils into two groups for each micronutrient (and how, even though it is mentioned in the supplement), 2. They investigated if bacterial and fungal communities differ between these two groups using the test ...?, 3. They selected

the bacterial and fungal genera that were significantly related to either the low or the high group. 4. They investigated the correlation between these selected genera and micronutrients.

L352: the correlations with bacteria were stronger than those with fungi.

L418: were not correlated

L437-438: The part of the sentence that states: 'on the basis that microbial communities were basically affected by climatic factor, land use, etc.' is apparently not related to the first part. Moreover, the results of Fig. S11 appear here for the first time (not mentioned in methods and results) so it is not clear why they are mentioned at this point. I suggest that this, and all other additional analyses that were done in response to reviewers' comments, are properly introduced in the method section so that it becomes evident what they represent.

L444: The soils in this study did not receive any specific micronutrients. My previous comment referred to other studies that examined the effect of micronutrient additions/fertilization on soil microbes (e.g. 'Microbial mechanism of zinc fertilizer input on rice grain yield and zinc content of polished rice' and others). I believe that mentioning the results of some of these studies and linking them to the results of this study could further improve the discussion section.

L 454: This study also showed the unique contribution of micronutrients when controlling for soil physicochemical properties, even though pH was correlated with soil micronutrients to some extent. This is not in any contradiction with the current study.

L 546: Here, the sentence could be slightly rephrased, like: 'The findings suggest that higher micronutrient concentrations could benefit plant productivity by increasing soil bacterial biomass', otherwise it seems like there is clear proof of the effect of micronutrients on microbial biomass and plant productivity in this study. Moreover, it would be good to separate (and elaborate on) the second part of this sentence to make two shorter sentences.

Fig. 2 The term 'interaction' was not replaced in this figure.

Reviewer #3 (Remarks to the Author):

The author supplemented the sampling process as required, added incubation experiments,

optimized the SEM models, and conducted metagenomic sequencing to correlate species and functions. In general, the authors have substantially improved the quality of their manuscript. I do have some minor suggestions for edits:

1. Although you have added the distribution of 180 soil samples, it is recommended to indicate the number of samples for each type of soil in the legend (Fig. S1).
2. L161-163: please use the units of volume and concentration correctly.
3. L186-189: Please modify the description of the qPCR process. Based on your description, only 40 cycles were performed in the second step, it seems incorrect.
4. L200: please check the references of the software used.
5. Please explain the sequencing depth of metagenomic data.
6. L202: please supplement the step size during assembly or display the entire k-mer list.
7. L206, L209: please check the standardization of expression of e-value cutoff.
8. Please check the image format of the attached figure, many of which have black lines at the edges.
9. L319: need to add spaces between parentheses, please check for such errors in full manuscript.
10. L388-391: This description is not very logical and should no longer explain the species origin of genes from the perspective of the overall community gene set. Here, we only need to explain the representative genes reported earlier from which species they originate, and then describe the abundance of these species in the community.
11. Please add the version of the software and database used in the material method.

12. Please add Fig. 5 resolution, while appropriately increasing the font size in the SEM models images (Fig. S5, Fig. S7, Fig. S10).

13. Please provide information on metagenome data availability in the manuscript (platform and ID for uploading metagenome data).

REVIEWER COMMENTS

Reviewer #1 (Remarks to the Author):

I read the response to my comments which were for the most part addressed. I also note the considerable revisions done in response to the other reviewers. There are some remaining questions.

Regarding point 3 in the responses to my comments, can the authors clarify what they mean by a "site"? Is it an area of a specific size, such as 10 x 10 m or something similar? Were there any criteria for choosing sites (such as avoiding certain features)? If this was described elsewhere, can this be shown in section 2.1?

Authors: We collected samples in an area of a specific size with 15 m * 15 m. We choose a site in a central land use area where avoid the disturbance of other land use and also avoid certain features such as road. Please see the Line 122-125.

Regarding point 4: I note that the authors included soil moisture in the SEM analysis, but how was soil moisture quantified?

Authors: Thank you for the good comment. Soil moisture was collected using the parameter of available water storage capacity from the HWSD (Harmonized World Soil Database v 1.2). In that database, this parameter is divided into seven classes. Thus, we used this moisture parameter as the categorical exogenous variable in the SEM analysis. Please see the description in Line 325-326.

Regarding the point about rarefaction and microbial alpha diversity (lines 251 and 252 of the revised manuscript): To justify their approach, the authors use a reference from 2011 which is key to the method, but the discussion about rarefying or not prior to alpha diversity analyses developed significantly since then. To my understanding, it is still not advised to rarefy prior to alpha diversity calculation. Some references discussing this include 10.3389/fmicb.2019.02407 and 10.1016/j.csbj.2021.12.036. If the authors can justify their approach with a recent and relevant reference, I am happy with it, otherwise, I would advise re-calculation of alpha diversity using an approach that reduces biases associated with rarefaction, for example by pooling samples as done in 10.1016/j.csbj.2021.12.036.

Authors: We are thankful the reviewer remind us of this controversy. After we carefully read these references, we agree with the reviewer that it is appropriate to calculate the alpha diversity before rarefaction. Thus, we have re-calculated the data of alpha diversity prior to rarefaction, but we found very small changes between two calculations. Now, the data in the Fig. 2, Fig. 4 and Fig. S10 have all been updated. In the text, we have changed the word "by" to "before" in the manuscript. Please see the Line 256.

Reviewer #2 (Remarks to the Author):

The authors performed a thorough revision of the manuscript with detailed responses to all the major concerns and they included additional data and analyses. Particular compliments to the authors for conducting a small incubation study that evaluated the influence of Fe and Zn additions on microbial community composition and functional genes.

Below are several additional points that the authors could take into account to clarify some remaining issues.

L39-40: It is unclear what happens with increasing micronutrients. The incubation experiment could receive a clearer introduction. For instance, the authors could state that they conducted an experiment with Zn and Fe additions for five different soil types showing that these micronutrients can indeed affect microbial community composition and functioning.

Authors: Thanks for the suggestion. Please see the revision in Line 40-42.

L92: are associated with productivity

Authors: The “production” has been replaced by “productivity”. Please see Line 93.

L96: to predict ecosystem responses to...? The sentence is not finished.

Authors: The “environmental changes” has been added. Please see Line 97.

L104: It is still not entirely clear here (and elsewhere) what the term ‘connection’ refers to without adding the context. It could be ‘fungal and bacterial network connectivity’ and later, for short just ‘network connectivity’.

Authors: All the “connection” have been replaced by “connectivity”. Please see the revision in Line 77-78, Line 104 and other revisions throughout the manuscript.

L 109-110: Perhaps a clearer version of this sentence would be something like: ‘... that micronutrients can explain a unique portion of the variation in soil microbiome structure and function in addition to pH and macronutrients’. In any case: the word ‘except’ here suggests that pH and macronutrients were not considered.

Authors: The sentence has been revised. Please see Line 109-111.

L 140: Could the authors add a reference to the method for the analyses of total micronutrients?

Authors: The reference has been added. Please see Line 142.

L 197: ‘to find’ or perhaps better ‘examine’

Authors: The “find” has been replaced by “examine”. Please see Line 199.

L 254: Could the authors clarify here that bacterial and fungal co-occurrence networks were constructed separately? Otherwise, it could be interpreted as a mixed microbial network.

Authors: The separate co-occurrence networks have been clarified. Please see Line 276.

L 260: This is not entirely clear. It could be stated that a higher network degree and average path length represent higher network connectivity.

Authors: This sentence has been revised. Please see Line 280-282.

L 261-269: This part requires some attention. It is not evident to what these taxa are correlated (L 262) (to each other, to micronutrients,... or were they ‘indicators’ of soils with low or high micronutrient concentrations?). What was the criterion for dividing the soils into these two groups

and which test was used to determine the difference in microbial community composition? I suggest that the authors consider the following order of description of the steps: 1. They divided the soils into two groups for each micronutrient (and how, even though it is mentioned in the supplement), 2. They investigated if bacterial and fungal communities differ between these two groups using the test ...?, 3. They selected the bacterial and fungal genera that were significantly related to either the low or the high group. 4. They investigated the correlation between these selected genera and micronutrients.

Authors: Thank you for the constructive suggestion. We have reorganized this part by using the order of description of the steps suggested by reviewer. Please Line 259-267.

L352: the correlations with bacteria were stronger than those with fungi.

Authors: The sentence has been revised. Please see Line 364-365.

L418: were not correlated

Authors: Please see the revision in Line 432-433.

L437-438: The part of the sentence that states: ‘on the basis that microbial communities were basically affected by climatic factor, land use, etc.’ is apparently not related to the first part. Moreover, the results of Fig. S11 appear here for the first time (not mentioned in methods and results) so it is not clear why they are mentioned at this point. I suggest that this, and all other additional analyses that were done in response to reviewers’ comments, are properly introduced in the method section so that it becomes evident what they represent.

Authors: The sentence has been removed and re-added in the result section. Please see Line 451 and Line 362-363. All the additional analyses required by reviewers are introduced in the method section. Please see Line 259-261 and others. The order of the figures in the supporting information has also been adjusted.

L444: The soils in this study did not receive any specific micronutrients. My previous comment referred to other studies that examined the effect of micronutrient additions/fertilization on soil microbes (e.g. ‘Microbial mechanism of zinc fertilizer input on rice grain yield and zinc content of polished rice’ and others). I believe that mentioning the results of some of these studies and linking them to the results of this study could further improve the discussion section.

Authors: Thank you for the suggestion. We have added two recent studies to link them to the results of our study and give some discussion. Please see Line 459-462.

L 454: This study also showed the unique contribution of micronutrients when controlling for soil physicochemical properties, even though pH was correlated with soil micronutrients to some extent. This is not in any contradiction with the current study.

Authors: Thank you for the comment. We have deleted this sentence to avoid the dispute. Please see Line 468. We also cited this reference in the other part. Please see Line 479-480.

L 546: Here, the sentence could be slightly rephrased, like: ‘The findings suggest that higher micronutrient concentrations could benefit plant productivity by increasing soil bacterial biomass’, otherwise it seems like there is clear proof of the effect of micronutrients on microbial biomass

and plant productivity in this study. Moreover, it would be good to separate (and elaborate on) the second part of this sentence to make two shorter sentences.

Authors: Thank you for the comments. We have rephrased the sentence and also divided into two shorter sentences. Please see Line 560-563.

Fig. 2 The term ‘interaction’ was not replaced in this figure.

Authors: The term “interaction” in Fig. 2 has been replaced by “network connectivity”. Please see new Fig. 2.

Reviewer #3 (Remarks to the Author):

The author supplemented the sampling process as required, added incubation experiments, optimized the SEM models, and conducted metagenomic sequencing to correlate species and functions. In general, the authors have substantially improved the quality of their manuscript. I do have some minor suggestions for edits:

1. Although you have added the distribution of 180 soil samples, it is recommended to indicate the number of samples for each type of soil in the legend (Fig. S1).

Authors: The number of samples for each type of soil has been added in the Fig. S1. Please see Fig. S1.

2. L161-163: please use the units of volume and concentration correctly.

Authors: The units of volume and concentration have been revised. Please see Line 162-165.

3. L186-189: Please modify the description of the qPCR process. Based on your description, only 40 cycles were performed in the second step, it seems incorrect.

Authors: Please see the revision in Line 189-191.

4. L200: please check the references of the software used.

Authors: The reference has been added. Please see Line 203.

5. Please explain the sequencing depth of metagenomic data.

Authors: The sequencing depth of metagenomic data was 20G. Please see Line 202.

6. L202: please supplement the step size during assembly or display the entire k-mer list.

Authors: The step size is 10. Please see the revision in Line 205.

7. L206, L209: please check the standardization of expression of e-value cutoff.

Authors: The standardization of expression of e-value cutoff has been added. Please see Line 210 and Line 213.

8. Please check the image format of the attached figure, many of which have black lines at the edges.

Authors: All the figures have been checked to avoid the black lines at the edges.

9. L319: need to add spaces between parentheses, please check for such errors in full manuscript.

Authors: The spaces have been added. Please see Line 330. Other errors have also been checked.

10. L388-391: This description is not very logical and should no longer explain the species origin of genes from the perspective of the overall community gene set. Here, we only need to explain the representative genes reported earlier from which species they originate, and then describe the abundance of these species in the community.

Authors: The sentences have been revised. Please see Line 401-405 and Table S8.

11. Please add the version of the software and database used in the material method.

Authors: The version of the software and database have been added. Please see Line 172, Line 178, Line 202, Line 204, Line 208, Line 209, Line 243, etc.

12. Please add Fig. 5 resolution, while appropriately increasing the font size in the SEM models images (Fig. S5, Fig. S7, Fig. S10).

Authors: The fig. 5 resolution has been added and the relevant font sizes have been enlarged.

13. Please provide information on metagenome data availability in the manuscript (platform and ID for uploading metagenome data).

Authors: The platform and ID have been added. Please see Line 213-214.